# STRIPAK directs PP2A activity toward MAP4K4 to promote oncogenic transformation of human cells

Jong Wook Kim[1,2,3,4†], Christian Berrios[2,5†], Miju Kim[1,2†], Amy E Schade[2,5†], Guillaume Adelmant[6,7,8], Huwate Yeerna[3], Emily Damato[1], Amanda Balboni Iniguez[1,9], Laurence Florens[10], Michael P Washburn[10,11], Kim Stegmaier[1,9], Nathanael S Gray[6], Pablo Tamayo[3,4], Ole Gjoerup[2], Jarrod A Marto[6,7,8], James DeCaprio[2,5,12]*, William C Hahn[1,2,12]*

[1]Broad Institute of Harvard and MIT, Cambridge, United States; [2]Department of Medical Oncology, Dana-Farber Cancer Institute, Boston, United States; [3]Division of Medical Genetics, School of Medicine, University of California, San Diego, San Diego, United States; [4]Moores Cancer Center, University of California, San Diego, San Diego, United States; [5]Program in Virology, Graduate School of Arts and Sciences, Harvard University, Cambridge, United States; [6]Department of Cancer Biology and Blais Proteomics Center, Dana-Farber Cancer Institute, Boston, United States; [7]Department of Pathology, Brigham and Women's Hospital and Harvard Medical School, Boston, United States; [8]Department of Oncologic Pathology, Dana-Farber Cancer Institute, Boston, United States; [9]Department of Pediatric Oncology, Dana-Farber Cancer Institute, Boston, United States; [10]Stowers Institute for Medical Research, Kansas City, United States; [11]Department of Pathology and Laboratory Medicine, University of Kansas Medical Center, Kansas City, United States; [12]Department of Medicine, Brigham and Women's Hospital and Harvard Medical School, Boston, United States

*For correspondence:
james_decaprio@dfci.harvard.edu (JDC);
william_hahn@dfci.harvard.edu (WCH)

†These authors contributed equally to this work

**Abstract** Alterations involving serine-threonine phosphatase PP2A subunits occur in a range of human cancers, and partial loss of PP2A function contributes to cell transformation. Displacement of regulatory B subunits by the SV40 Small T antigen (ST) or mutation/deletion of PP2A subunits alters the abundance and types of PP2A complexes in cells, leading to transformation. Here, we show that ST not only displaces common PP2A B subunits but also promotes A-C subunit interactions with alternative B subunits (B''', striatins) that are components of the Striatin-interacting phosphatase and kinase (STRIPAK) complex. We found that STRN4, a member of STRIPAK, is associated with ST and is required for ST-PP2A-induced cell transformation. ST recruitment of STRIPAK facilitates PP2A-mediated dephosphorylation of MAP4K4 and induces cell transformation through the activation of the Hippo pathway effector YAP1. These observations identify an unanticipated role of MAP4K4 in transformation and show that the STRIPAK complex regulates PP2A specificity and activity.

## Introduction

Protein phosphorylation plays a regulatory role in nearly all biological processes and dysregulation of protein phosphorylation contributes to many diseases. Both kinases and phosphatases have been implicated in the pathogenesis of specific cancers, and several small molecule kinase inhibitors are

**eLife digest** Cells maintain a fine balance of signals that promote or counter cell growth and division. Two sets of enzymes – called kinases and phosphatases – contribute to this balance. In general, kinases "switch on" other proteins by tagging them with a phosphate molecule. This process is called phosphorylation. Phosphatases, on the other hand, dephosphorylate these proteins, switching them off. Cancer cells often have mutations that activate kinases to drive cancer growth. The same cells can have mutations that inactivate the phosphatases or reduce their abundance. The roles of phosphatases in cancer are still being studied. One major hurdle in this research is that it is not always clear how they recognize the proteins they dephosphorylate.

Protein phosphatase 2A (or PP2A for short) is one of the phosphatases that is often mutated or deleted in human cancers. Even just reduced levels of PP2A can promote cancer. Kim, Berrios, Kim, Schade et al. used an experimental trick to decrease the phosphatase activity of PP2A in human cells growing in a dish. Biochemical analysis of these cells showed that, as expected, many proteins were now in their phosphorylated states. Unexpectedly, however, some proteins were dephosphorylated under these conditions. One of these proteins was called MAP4K4. In the case of MAP4K4, the dephosphorylated state contributes to the growth of the cancer cell. Kim et al. carried out further genetic and biochemical experiments to show that, in these cells, PP2A and MAP4K4 stay physically connected to one another. This connection was enabled by a group of proteins called the STRIPAK complex. The STRIPAK proteins directed the remaining PP2A towards MAP4K4. Low levels or activity of PP2A could, therefore, promote cancer in a different way.

Taken together, PP2A is not a single phosphatase that always turns proteins off, but rather is a dual switch that turns off some proteins while turning on others. Future experiments will explore to what extent these findings also apply in tumors. Information about how mutations in PP2A affect human cancers could suggest new targets for cancer drugs.

standard treatments in such cancers. In addition, several phosphatases have been identified as tumor suppressors (*Sablina and Hahn, 2007*; *Lawrence et al., 2014*).

PP2A, an abundant serine/threonine phosphatase in mammalian cells, is comprised of three subunits: A (structural), B (regulatory), and C (catalytic). The A and C subunits form the core enzyme and interact with different B regulatory subunits to create many distinct PP2A enzymes (*Pallas et al., 1990*; *Chen et al., 2007*; *Cho et al., 2007*; *Shi, 2009*; *Sents et al., 2013*). Moreover, there are two A and two C isoforms, and at least four classes of B subunits B, B', B'', and B''' (striatins), each of which exist as several different isoforms. Although the prevailing view is that the B subunits provide substrate specificity, how B subunits accomplish this regulation remains unclear (*Shi, 2009*; *Hertz et al., 2016*).

Genome characterization studies of human cancers have identified recurrent mutations and deletions involving PP2A subunits. Indeed, the PP2A Aα (*PPP2R1A*) subunit ranks among the most recurrently mutated gene across many cancer types (*Lawrence et al., 2014*). Notably, mutations in Aα occur at high frequency in premalignant endometrial lesions (*Anglesio et al., 2017*). PP2A is also a target of the Small T antigens (ST) of SV40 and other polyomaviruses including the human oncogenic Merkel cell polyomavirus (*Pallas et al., 1990*; *Chen et al., 2004*; *Cheng et al., 2017*), and this interaction contributes to cell transformation (*Hahn et al., 2002*). Structural studies have shown that ST disrupts the formation of a functional PP2A holoenzyme by displacing or hindering B subunit access to the PP2A core-enzyme (*Chen et al., 2007*; *Cho et al., 2007*). However, ST has a lower binding affinity in vitro for the PP2A core enzyme than B' subunits, which suggests that ST interaction with the core enzyme may either occur prior to the B subunit binding or ST directly inhibits PP2A activity independently of subunit assembly (*Chen et al., 2007*).

Several investigators have used mass spectrometry to identify proteins that interact with PP2A (*Goudreault et al., 2009*; *Herzog et al., 2012*). These studies identified a large complex called the Striatin-interacting phosphatase and kinase (STRIPAK) complex (*Goudreault et al., 2009*). The STRIPAK complex contains striatin family (STRN) proteins, several kinases, scaffolding proteins, and PP2A subunits. Indeed, striatins were initially described as non-canonical PP2A regulatory subunits (B''' subunits) (*Moreno et al., 2000*). STRIPAK complexes have also been shown to associate with

members of the GCKIII kinase subfamily (*MST3, STK24, and STK25*) (*Kean et al., 2011*). In addition, mitogen-activated protein kinase kinase kinase kinase 4 (*MAP4K4*), a Ste20-like kinase, although not an obligate member of the STRIPAK complex, associates with STRIPAK (*Frost et al., 2012*; *Herzog et al., 2012*; *Hyodo et al., 2012*). We also identified members of the STRIPAK complex, including STRN3, STRN4, STRIP1, and MAP4K4 in complex with SV40 ST (*Rozenblatt-Rosen et al., 2012*). Although STRIPAK comprises multiple signaling enzymes, it is unclear how disruptions to the biochemical complex integrate with or disrupt phosphorylation cascades; or whether these signaling alterations synergize with ST to mediate cellular transformation.

*MAP4K4* is a serine/threonine kinase that was initially found to activate the c-Jun N-terminal kinase (JNK) signaling pathway (*Yao et al., 1999*), downstream of TNF-α. *MAP4K4* has also been implicated in a large number of biological processes including insulin resistance, focal adhesion disassembly, as well as cellular invasion and migration (*Collins et al., 2006*; *Tang et al., 2006*; *Yue et al., 2014*; *Danai et al., 2015*; *Vitorino et al., 2015*). Recent studies have shown that MAP4K4 phosphorylates LATS1/2, activating the Hippo tumor suppressor pathway, leading to YAP1 inactivation (*Mohseni et al., 2014*; *Meng et al., 2015*; *Zheng et al., 2015*). Here, we investigated the role of the STRIPAK complex and *MAP4K4* in human cell transformation driven by SV40 ST and found that kinase inactivation or partial suppression of *MAP4K4* replace the expression of ST in the transformation of human cells.

## Results

### Identification of MAP4K4 as a candidate phosphoprotein targeted in cells transformed by PP2A perturbation

Human embryonic kidney (HEK) epithelial cells expressing SV40 Large T antigen (LT), the telomerase catalytic subunit (*hTERT*), and oncogenic HRAS (referred to as HEK TER hereafter) have served as a useful model system to identify pathways and protein complexes that can functionally substitute for SV40 ST in promoting transformation, including partial depletion of PP2A (*Chen et al., 2004*; *Sablina et al., 2010*). These cells, upon expression of SV40 ST or partial knockdown of PP2A Aα or Cα subunits, become tumorigenic (*Hahn et al., 2002*; *Chen et al., 2004*). Prior studies have shown that expression of ST, or partial inhibition of certain PP2A subunits, causes increased phosphorylation of PP2A substrates (*Sablina and Hahn, 2008*; *Sablina et al., 2010*).

To assess the serine/threonine phosphorylation events that are associated with transformation induced by ST or by partial knockdown of PP2A, we performed global Isobaric Tags for Relative and Absolute Quantitation (iTRAQ) phosphoproteomic profiling of HEK TER cells expressing ST (HEK TER ST) or in which expression of the PP2A Aα, Cα, or B56γ subunits were fully or partially suppressed using previously characterized shRNAs (*Figure 1A*, *Figure 1—figure supplement 1A–B*) (*Sablina et al., 2010*). We also confirmed that these genetic perturbations promoted the transformation phenotype as gauged by anchorage-independent (AI) growth assays as previously described (*Figure 1—figure supplement 1C*) (*Sablina et al., 2010*). Through mass spectrometry analysis of the phosphopeptides altered across these conditions, we identified 6025 phosphopeptides corresponding to 2428 individual proteins reproducibly detected in two replicate experiments. Processing and normalization of the raw data were performed using corresponding control experiments (GFP control for ST, *shLuciferase* (*shLuc*) control for shRNAs against PP2A, see methods for details). We then performed comparative marker selection analysis (*Gould et al., 2006*) to identify candidate phosphoproteins that were most significantly correlated with the transformation phenotype (*Figure 1A*). In consonance with previous studies (*Ratcliffe et al., 2000*; *Kuo et al., 2008*), we observed an increase in phosphorylation of direct or indirect targets of PP2A, including AKT1S and β-catenin (*CTNNB1*) in cells which were transformed by either expressing ST or partial knockdown of PP2A Cα subunit in HEK TER cells (*Figure 1B*) (*Sablina et al., 2010*). Conversely, we also observed decreased phosphorylation on multiple proteins in cells transformed by ST or by PP2A perturbation (B56γ1, Cα2). Notably, the phosphorylation signature for transformation included four distinct sites on MAP4K4 (T804, S888, S889, S1272, p<0.05, *Figure 1B*).

Our previous systematic analysis of SV40 ST identified MAP4K4, in addition to PP2A and other STRIPAK components in the same complex (*STRN4, STRN3, CTTNBP2NL, FAM40A, MAP2K3, STK24, PPP2R1A*) (*Rozenblatt-Rosen et al., 2012*). To confirm these interactions, we generated

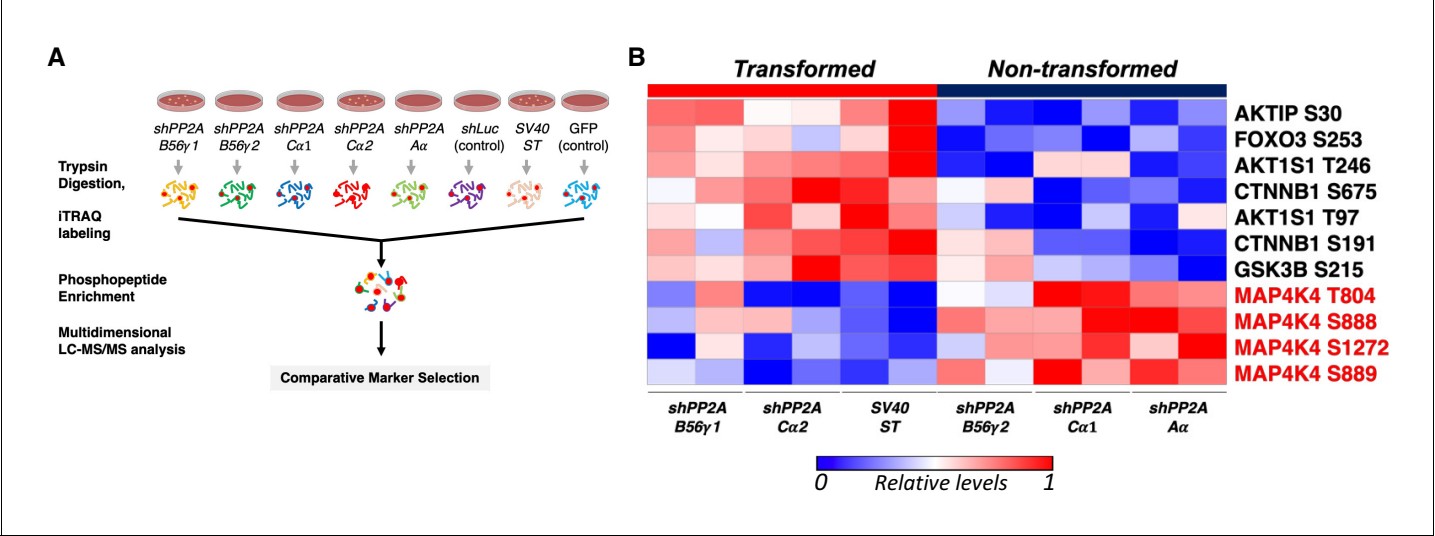

**Figure 1.** Global phosphoproteomic analysis identifies MAP4K4 dephosphorylation in cells transformed by PP2A perturbation. (**A**) Schematic illustrating the global phosphoproteomics experiment. (**B**) The heatmap depicts phosphopeptides that are either positively or negatively correlated with the transformation phenotype (p<0.05, FDR < 1). Each column represents individual samples that were normalized to *shLuc* for *shPP2A* or in the case of ST to GFP control. The sample designations after the normalization and comparative marker selection analysis are shown below the heatmap, with each sample shown in replicates. A selected subset of phosphorylated sites which distinguishes transforming and non-transforming perturbations are shown. The online version of this article includes the following figure supplement(s) for figure 1:

**Figure supplement 1.** Changes in PP2A levels and AI growth with PP2A knockdown and STRIPAK interactions with ST from HPyV.

lentiviral C-terminal Flag-HA Tandem Affinity Purification (CTAP) constructs for SV40 ST as well as ST from three closely related Human Polyoma Viruses (HPyV) including JCPyV-CY, JCPyV-Mad1, and BKPyV, along with GFP as a negative control. We introduced these viral proteins into HCT116 cells and performed HA-tag immunoprecipitations (IP) from lysates of cells expressing ST from the respective viruses. We confirmed co-complex formation between SV40 ST and MAP4K4, as well as with STRIPAK components PP2A C, STRN3, and STRIP1 (*Figure 1—figure supplement 1D*). We observed that ST of JCPyV and BKPyV, the two most closely related HPyVs to SV40, also interacted with STRN3, STRIP1, and PP2A C but not MAP4K4, indicating that the interaction of SV40 ST and MAP4K4 was unique to SV40 ST. The association of ST with B‴ subunits (striatins) was unexpected, because ST was previously reported to primarily bind PP2A Aα and displace most B subunits (*Pallas et al., 1990*; *Chen et al., 2007*; *Cho et al., 2007*; *Sablina et al., 2010*). These observations raised the possibility that ST modulates MAP4K4 phosphorylation via PP2A activity associated with the STRIPAK complex.

## Partial knockdown of MAP4K4 promotes cell transformation

To determine if MAP4K4 and other SV40 ST interacting proteins participated in cell transformation, we created and stably expressed two distinct shRNAs targeting each of several SV40 ST interacting proteins, including *STRN3, STRN4, STRIP1, MARCKS, MAP4K4, and STK24* in HEK TER cells. We then assessed the ability of each of these shRNAs to promote AI growth, a readout for the transformed phenotype (*Figure 2A*, *Figure 2—figure supplement 1A*). As expected, expression of SV40 ST or partial knockdown of PP2A Cα subunit in HEK TER cells induced robust AI growth (*Figure 2—figure supplement 1A*). Among the STRIPAK components, we found that one of the two shRNAs targeting *MAP4K4* (*shMAP4K4-82*) elicited a potent transformation phenotype (*Figure 2B–C*, *Figure 2—figure supplement 1A*). To ensure that the observed phenotype was specific to targeting MAP4K4 and not due to an off-target effect of this shRNA, we repeated the AI growth assay using eight different MAP4K4-targeting shRNAs including the two shRNAs used in the initial experiment (*Figure 2—figure supplement 1B*). In addition, we found that the three shRNAs which promoted HEK TER cells to grow in an AI manner (*shRNA-82, 92, 93*) only partially suppressed MAP4K4 levels (*Figure 2—figure supplement 1B*). Specifically, we focused on *shMAP4K4-82*, which promoted the

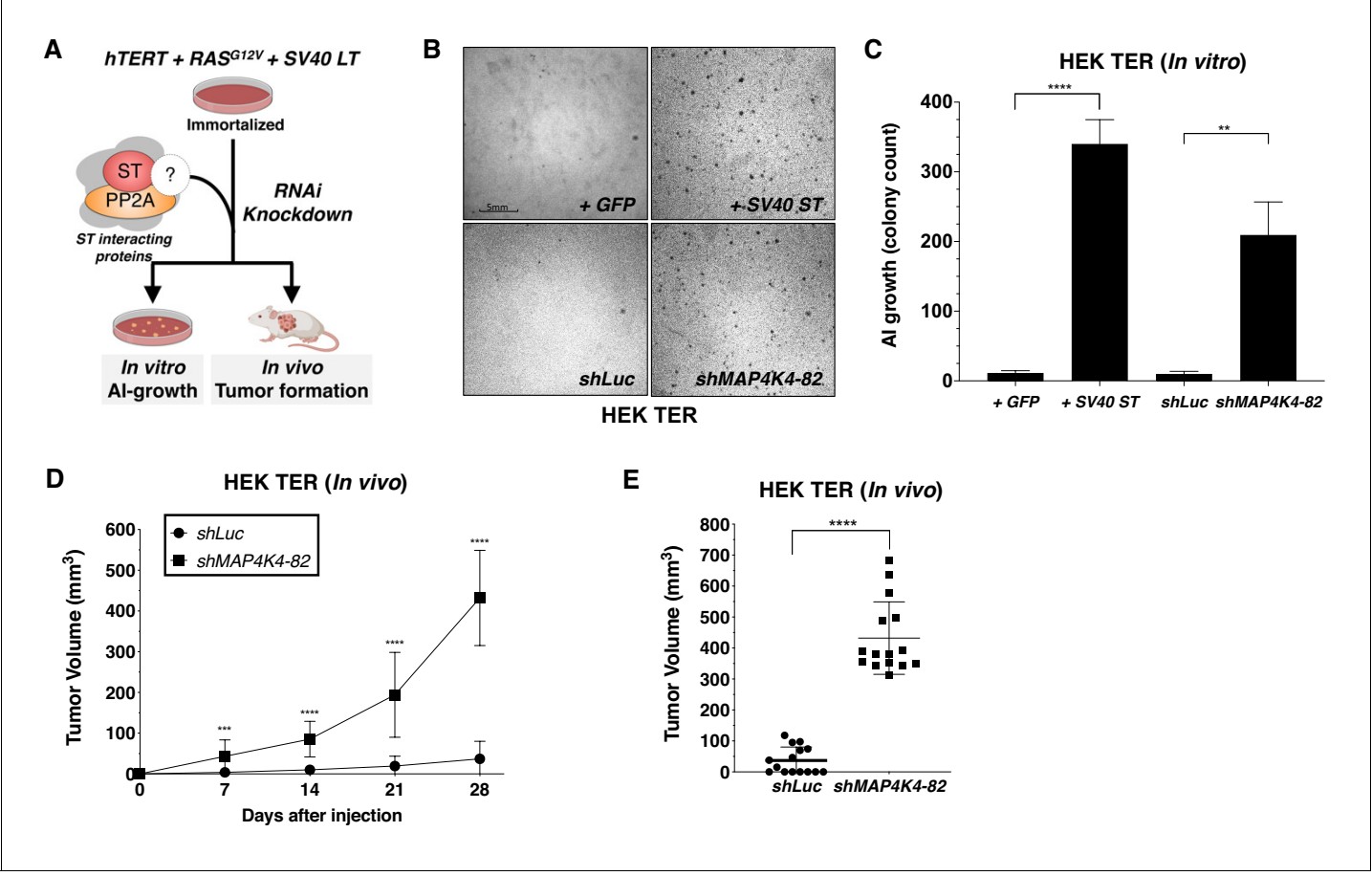

**Figure 2.** Partial knockdown of *MAP4K4* expression promotes oncogenic transformation and tumor formation. (A) Schematic of experimental design to reveal binding proteins that when depleted can substitute for ST in transformation (B) Representative image of AI growth induced by ST or *MAP4K4* partial knockdown (Grid shows 5 mm). (C) Quantification of AI growth following expression of *MAP4K4 shRNA-82* (*shMAP4K4*), SV40 ST or corresponding controls (GFP or *shLuc*). Graph depicting tumor volume as a function of time (D) or an endpoint at day 28 (E) for subcutaneous xenografts expressing *shLuc* control or *shMAP4K4-82* in HEK TER cells (Student's t-test, **p<0.001, ***p<0.0001, ****p<0.00001).
The online version of this article includes the following source data and figure supplement(s) for figure 2:

**Source data 1.** Quantification of soft-agar colony and tumor volume with MAP4K4 knockdown.
**Figure supplement 1.** Changes in AI growth with MAP4K4 knockdown.

most robust AI growth and knocked down *MAP4K4* mRNA levels by 50% (*Figure 2—figure supplement 1B–C*). In contrast, none of the shRNAs that induced more than 50% knockdown of *MAP4K4* expression resulted in AI growth (*Figure 2—figure supplement 1B*). This relationship between partial knockdown and cell transformation is similar to what has been reported for the knockdown of PP2A Aα and Cα subunits (*Chen et al., 2005*; *Sablina et al., 2010*). To further confirm these data in vivo, we performed xenograft experiments to assess tumor formation by subcutaneous injection of immunodeficient mice. Consistent with the in vitro studies, *shMAP4K4-82* induced potent tumor formation when compared to the *shLuc* control (*Figure 2D–E*). These observations suggest that partial knockdown, but not full depletion, of MAP4K4, promotes both transformation and tumor formation.

## SV40 ST promotes the interaction of MAP4K4 with STRIPAK

To understand the mechanism by which the ST/MAP4K4 axis contributes to cell transformation, we first assessed changes in interactions between MAP4K4 and its binding partners upon ST expression. Specifically, we stably expressed NTAP-MAP4K4 in HEK TER cells expressing either ST or GFP as a negative control. We used Stable Isotope Labeling with Amino Acids (SILAC) to encode proteins in each condition (*Figure 3A*). We found that MAP4K4 interacted with STRIPAK components, including

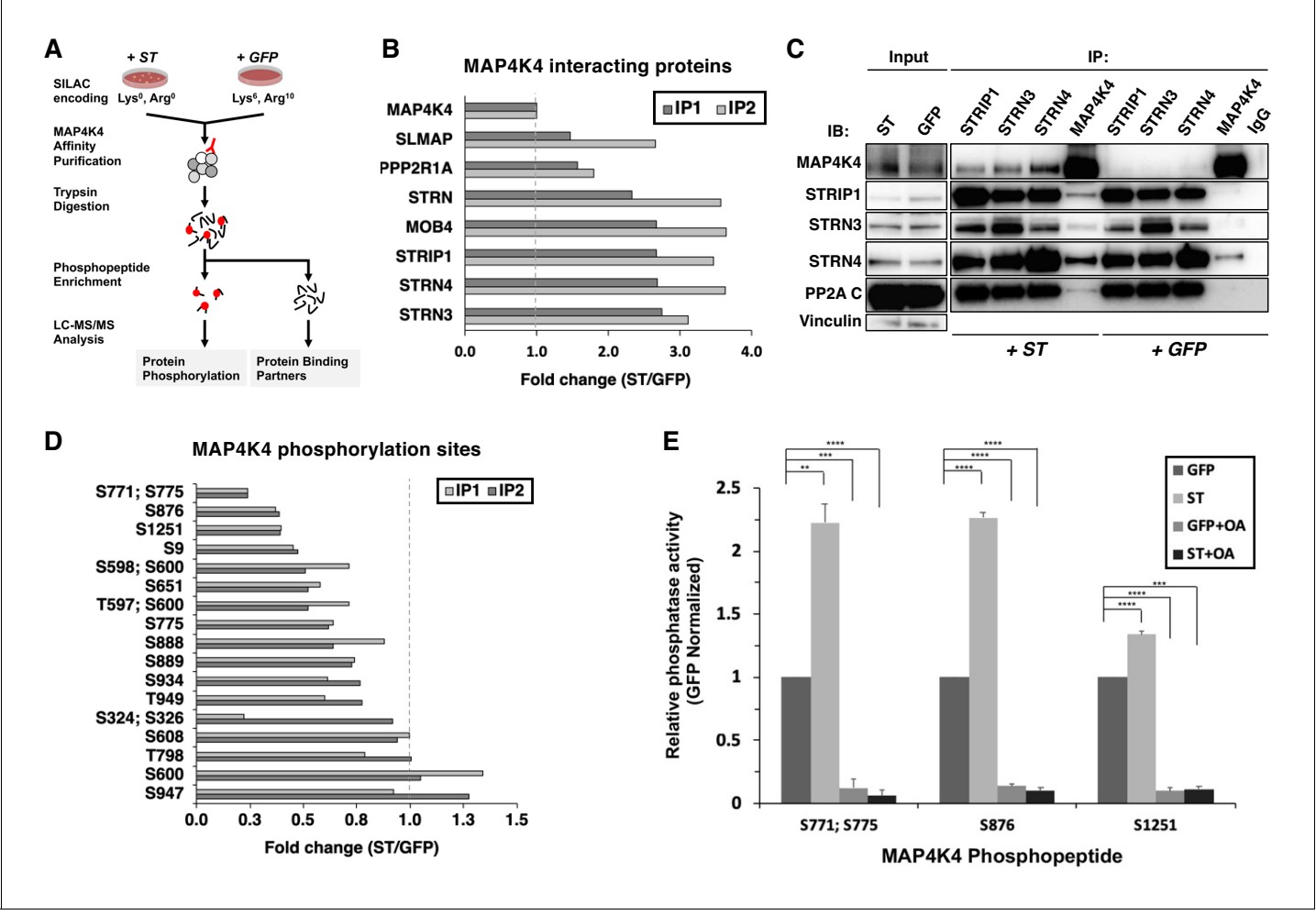

**Figure 3.** ST promotes MAP4K4 interactions with STRIPAK and MAP4K4 dephosphorylation. (**A**) Schematic of targeted proteomic analysis of MAP4K4 phosphorylation and interacting proteins in the presence of *ST* or *GFP* control. (**B**) SILAC experiment in which MAP4K4-associated proteins were assessed in cells expressing ST or a GFP control by SILAC experiments performed as biological replicates (IP1, IP2). The proteins that showed a fold change above one have increased interactions with MAP4K4 in ST expressing cells relative to GFP. All values of the retrieved peptides were normalized to the total number of MAP4K4 peptides, prior to calculating the ratios between ST- versus GFP-expressing cells to account for variations in the amount of MAP4K4 after affinity purification. (**C**) Immunoblot showing results of a Co-Immunoprecipitation (Co-IP) analysis of components of STRIPAK with MAP4K4 in ST- or GFP-expressing cells. ST induced the association of MAP4K4 with STRIPAK components. (**D**) Quantification of fold changes in the abundance of MAP4K4 phosphorylation across indicated sites (y-axis) in cells expressing ST relative to the GFP control in two independent experiments (IP1, IP2). The phosphosites with fold changes below one show a decrease of phosphorylation of MAP4K4 in ST expressing cells relative to GFP. All values of the retrieved peptides were normalized to the total number of MAP4K4 peptides, prior to calculating the ratios between ST- versus GFP-expressing cells to account for variations in the amount of MAP4K4 after affinity purification. (**E**) After immunoprecipitation of STRN4 from ST- versus GFP-expressing cells, in vitro PP2A activity was measured with synthetic MAP4K4 peptides (S771;S775, S876, or S1251) identified in the targeted phosphoproteomic experiments (x-axis). Relative phosphatase activity in ST- relative to GFP-expressing HEK TER cells is shown for each phosphopeptide (y-axis). Okadaic acid (OA) was used to inhibit PP2A activity in parallel conditions (Student's t-test, **p<0.001, ***p<0.0001, ****p<0.00001).

The online version of this article includes the following source data and figure supplement(s) for figure 3:

**Source data 1.** Quantification of MAP4K4 interacting proteins and phosphopeptides.

**Figure supplement 1.** Changes in MAP4K4-STRIPAK interactions with ST expression.

STRIP1, STRN3, STRN4, and the PP2A Aα subunit. The interactions between MAP4K4 and the STRI-PAK components were increased by 3–4-fold in cells expressing ST relative to the GFP control (*Figure 3B*). We tested a series of ST mutants (R21A, W147A, F148A, P132A) that are unable to bind to PP2A Aα (*Cho et al., 2007*) in 293 T cells, and found that these mutant ST proteins were unable to interact with STRN3, a core component of the STRIPAK complex (*Figure 3—figure*

*supplement 1A*), demonstrating that this interaction is dependent on ST binding to PP2A Aα subunit. These observations indicated that MAP4K4 interaction with STRIPAK is enhanced in cells expressing SV40 ST.

To corroborate these observations, we performed IP of endogenous STRN3, STRN4, STRIP1, and MAP4K4 and compared the interactions of components of the STRIPAK complex with MAP4K4 in HEK TER cells expressing either ST or GFP. Consistent with the proteomic results, we observed that the interaction of MAP4K4 with the STRIPAK complex was significantly enhanced in the presence of ST (*Figure 3C*). We also performed these experiments in normal human fibroblasts (IMR90) expressing ST or GFP (negative control) and confirmed the enhanced binding of MAP4K4 to STRN4 and STRIP1 (*Figure 3—figure supplement 1B*). These observations indicate that interactions between MAP4K4 and STRIPAK components, including STRIP1, STRN3, and STRN4 are enhanced in the presence of SV40 ST.

We next analyzed the enriched phosphopeptides from affinity-purified MAP4K4 (*Figure 3A*) to better interrogate the full phosphorylation landscape on the kinase. In two independent experiments, we quantified 17 MAP4K4 phosphorylation sites (*Figure 3D*). The majority of these sites exhibited reduced phosphorylation in cells expressing ST. These findings further demonstrate that ST mediates dephosphorylation of several distinct MAP4K4 sites.

To evaluate if MAP4K4 dephosphorylation is mediated by the STRIPAK complex, we isolated STRN4 from cells expressing ST or GFP (*Figure 3—figure supplement 1C*) and measured PP2A-specific dephosphorylation activity using synthetic phosphopeptides encompassing MAP4K4 sites S771; S775, S876, or S1251. We selected these sites because they exhibited the largest change in phosphorylation upon ST expression (*Figure 3D*). As a control, we treated parallel samples with okadaic acid (OA), a potent and specific PP2A inhibitor. As expected, we observed that OA treatment eliminated phosphatase activity under all conditions (*Figure 3E*). In contrast, co-incubation of MAP4K4 phosphopeptides with STRN4 immune complexes from ST-expressing cells led to dephosphorylation of the S771/S775 and S876 phosphopeptides by greater than twofold compared to GFP control, while we found a modest but reproducible increase of dephosphorylation of the S1251 site (*Figure 3E*). These observations suggest that ST promotes PP2A-mediated dephosphorylation of MAP4K4 in the STRN4 complex.

## Attenuation of MAP4K4 kinase activity is associated with transformation

Since MAP4K4 phosphorylation at several sites was substantially attenuated in the presence of ST, we assessed whether this decrease in MAP4K4 phosphorylation affected MAP4K4 activity by performing an in vitro kinase assay using tandem-affinity purified MAP4K4 from cells that expressed ST or GFP control. We found that the activity of MAP4K4 was reduced in ST-expressing cells compared to cells that expressed GFP control (*Figure 4A*, *Figure 4—figure supplement 1A*).

To assess the relevance of MAP4K4 kinase activity to the transformation phenotype, we tested the consequences of pharmacological or genetic inhibition of MAP4K4 on AI growth. Specifically, we treated HEK TER cells with a previously described small molecule inhibitor of MAP4K4 (compound 29) (*Crawford et al., 2014*) over a range of concentrations (0–2 µM) and assessed MAP4K4 activity (*Figure 4—figure supplement 1B*) and AI cell growth (*Figure 4B*). In consonance with what we observed with partially knocked down MAP4K4 expression, escalating doses of this MAP4K4 inhibitor led to an increase in the number of AI colonies until it reached 2 µM when MAP4K4 kinase activity was inhibited more than 90% as measured by in vitro kinase assays (2 µM, *Figure 4B*, *Figure 4—figure supplement 1A*). We found that compound 29 induced modest effects on cell proliferation over the range of tested concentrations (*Figure 4—figure supplement 1C*). Consistent with the results from the genetic experiments (*Figure 2B–C*), we observed that partial inhibition of MAP4K4 activity led to increased AI growth.

We also tested whether inhibiting MAP4K4 by expressing a loss-of-function MAP4K4 allele promoted transformation. The kinasedead MAP4K4 K54R allele has previously been demonstrated to act as a dominant interfering mutant (*Wang et al., 2013*). We created HEK TER cells stably expressing kinase-dead (K54R) or the wild-type (WT) version of MAP4K4 and confirmed the loss of kinase activity for the MAP4K4 mutant allele (*Figure 4—figure supplement 1D–E*). When we performed AI growth assays, we observed that the introduction of MAP4K4 K54R but not WT MAP4K4 induced cell transformation (*Figure 4C*). Together, these observations demonstrate that partial depletion or

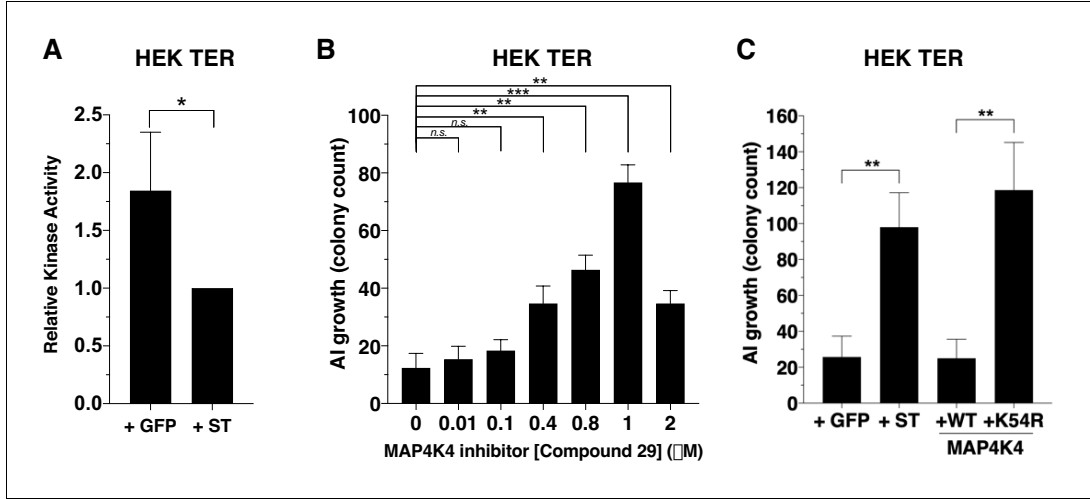

**Figure 4.** Partial inhibition of *MAP4K4* kinase activity elicits transformation. (**A**) Quantification of MAP4K4 in vitro kinase activity after MAP4K4 was tandem-affinity purified from cells expressing ST or GFP control. (**B**) Quantification of AI growth after increasing concentrations of the MAP4K4 inhibitor C29. (**C**) Quantification of AI growth after expression of MAP4K4 WT, MAP4K4 K54R *mutant*, ST, or GFP in HEK TER cells (Student's t-test, *p<0.01, **p<0.001, ***p<0.0001, n.s. = not significant).

The online version of this article includes the following source data and figure supplement(s) for figure 4:

**Source data 1.** Quantification of AI growth and in vitro MAP4K4 kinase activity.

**Figure supplement 1.** Changes in AI growth and proliferation with inhibition of MAP4K4 kinase activities.

inhibition of MAP4K4 activity mimics ST in inducing transformation and that attenuation of MAP4K4 kinase activity is associated with ST-induced cell transformation.

## STRN4 is required for ST-mediated transformation

Reduction of MAP4K4 levels and activity was sufficient to drive transformation in the absence of ST; therefore, we also investigated whether members of the STRIPAK complex were required for ST-mediated oncogenic transformation (*Figure 5A*). Specifically, we assessed the consequences of depleting components of STRIPAK in HER TER ST cell and found that knockdown of *STRN4* led to a significant reduction in transformation (*Figure 5B–C*). We tested 4 *STRN4*-targeting shRNAs and observed reduction in AI colonies in a manner that significantly correlated with the degree of *STRN4* knockdown (*Figure 5—figure supplement 1A–B*). To confirm that these findings were not due to an off-target effect of RNAi, we created a *STRN4* allele (*STRN4-58R*) resistant to the *STRN4*-specific shRNA (*shSTRN4-58*) and expressed this in HEK TER ST cells (*Figure 5—figure supplement 1C*). We found that expression of this *STRN4* allele rescued the effects of suppressing *STRN4* on AI growth (*Figure 5D*). We also deleted STRN4 using CRISPR-Cas9 gene editing and further confirmed that *STRN4* expression was required for ST-induced cell transformation (*Figure 5—figure supplement 1D*). We assessed the consequences of knocking down *STRN4* in vivo and found that *STRN4* knockdown significantly reduced tumor formation of HEK TER ST cells (*Figure 5E–F*). Collectively, these observations demonstrate that *STRN4* is required for ST-mediated transformation and tumor formation.

## STRN4 is required for the STRIPAK complex to associate with MAP4K4

To assess whether ST modulates interactions involving STRN4, we isolated endogenous STRN4 from cells expressing either ST or a GFP control and performed a proteomic analysis of associated proteins (*Figure 6A*). We found that STRN4 interactions with MAP4K4 were increased 1.6-fold, while interactions with the PP2A C subunits did not change, in cells expressing SV40 ST relative to GFP control (*Figure 6B*, *Figure 6—figure supplement 1A*).

Because ST promoted the interaction of MAP4K4 with STRN4, we evaluated the role of STRN4 in organizing the STRIPAK complex. When we assessed the impact of knocking down STRN4 on the

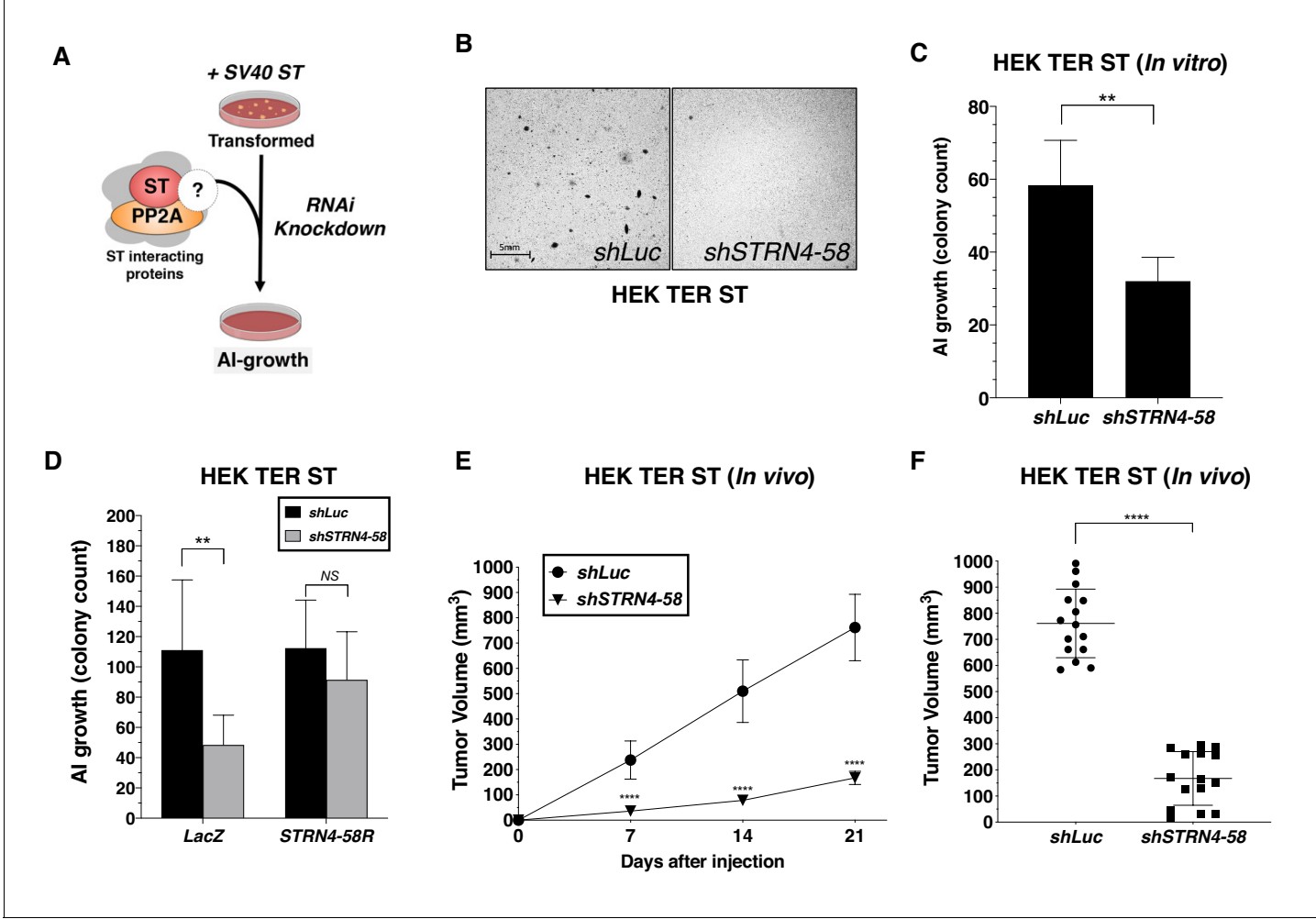

**Figure 5.** *STRN4* is required for ST-mediated transformation and tumor induction. (**A**) Schematic of experimental design to reveal binding proteins, that when depleted, inhibit ST-mediated transformation. (**B**) Representative images of AI colonies observed after knockdown of *STRN4* in HEK TER ST cells with *shSTRN4-58*. (**C**) Quantification of the number of AI colonies following the introduction of *shSTRN4-58* or *shLuc* control. (**D**) Quantification of the number of AI colonies after expression of *shSTRN4-58* in the presence (*STRN4-58R*) or absence (*LacZ*) of an shRNA-resistant *STRN4* cDNA. Tumor volume as a function of time (**E**) or at the endpoint at day 21 (**F**) for subcutaneous xenografts expressing shLuc control or *STRN4 shRNA* (*shSTRN4-58*) in HEK TER ST cells (Student's t-test, **p<0.001, ****p<0.00001).

The online version of this article includes the following source data and figure supplement(s) for figure 5:

**Source data 1.** Quantification of soft-agar colony and tumor volume with STRN4 knockdown.

**Figure supplement 1.** Changes in AI growth with STRN4 knockdown.

STRIPAK complex in HEK TER ST cells by Co-IP of endogenous STRIP1, STRN4, and MAP4K4 (***Figure 6C***, ***Figure 6—figure supplement 1B***) with or without STRN4 suppression, we observed that interactions of MAP4K4 with other members of STRIPAK (STRIP1, PP2A C) were attenuated when *STRN4* was suppressed, indicating that STRN4 is required for MAP4K4 interactions with the STRIPAK complex.

Prior studies have shown that Striatins act as scaffolds in the STRIPAK complex (***Chen et al., 2014a***). Based on these observations, we hypothesized that depletion of STRN4 in the presence of ST would lead to dissociation of MAP4K4 from the STRIPAK complex, which in turn would increase MAP4K4 activity. To test this hypothesis, we performed an in vitro kinase assay using MAP4K4 isolated from HEK TER ST cells expressing either control or *STRN4*-specific shRNA. We observed a modest, but statistically significant (p<0.05) increase in MAP4K4 kinase activity when *STRN4* was suppressed (***Figure 6D***, ***Figure 6—figure supplement 1C***). Moreover, we found that co-knockdown

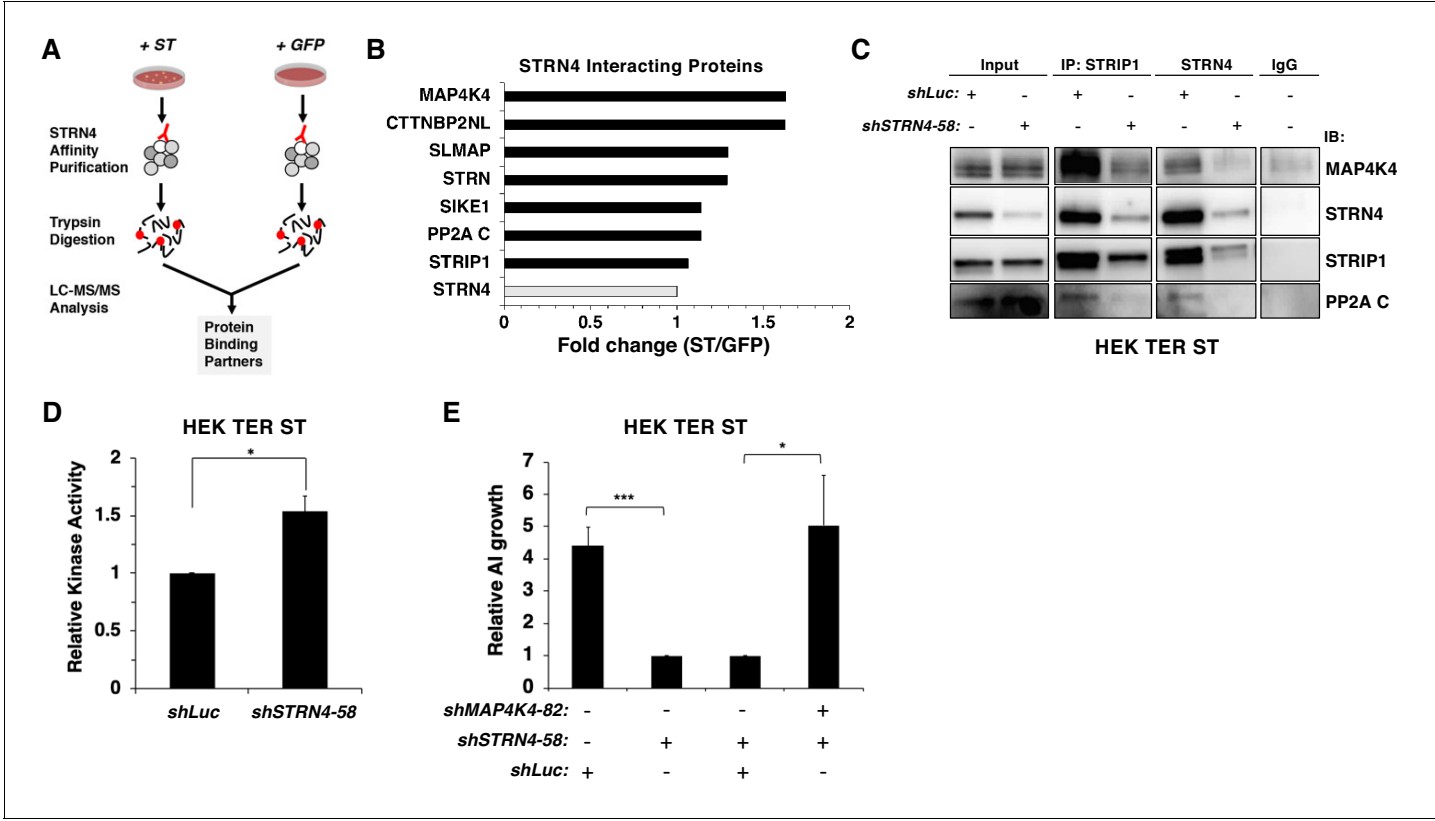

**Figure 6.** STRN4 is required for STRIPAK to interact with MAP4K4. (**A**) Schematic of proteomic analysis of STRN4 interacting proteins in the presence of ST or a GFP control. (**B**) Fold change in abundance of STRN4 interacting proteins in the presence of ST compared to the GFP control. All values were normalized to STRN4 levels to account for variations in the total amount of STRN4 isolated from cells expression ST or GFP control. Fold-change was calculated to reflect differences in the amount of STRN4 interacting proteins between cells expressing ST relative to GFP. Interactions of a number of proteins in the STRIPAK complex, including PP2A C, MAP4K4, CTTNBP2NL, SIKE1, and SLMAP with STRN4, were increased in ST-expressing cells relative to the GFP control. (**C**) Immunoblot showing a Co-IP analysis of STRIPAK core components and MAP4K4 after knockdown of *STRN4* using shRNA. STRN4 is required for the STRIPAK component STRIP1 to interact with MAP4K4 and PP2A C. (**D**) Quantification of MAP4K4 in vitro kinase activity after *STRN4* knockdown (*shSTRN4-58*). (**E**) Quantification of AI growth after expression of *STRN4* shRNA with or without co-expression of *MAP4K4 shRNA* in HEK TER ST cells. Suppression of *MAP4K4* expression rescued the transformation defect arising from *STRN4* depletion. All experiments were performed in triplicate, and the statistical analyses were performed relative to the controls (Student's t-test, *p<0.01, ***p<0.0001). The online version of this article includes the following source data and figure supplement(s) for figure 6:

**Source data 1.** Qunatification of STRN4 interacting proteins and in vitro MAP4K4 kinase activity and AI growth.
**Figure supplement 1.** Changes in MAP4K4 kinase activity and STRIPAK interactions with STRN4 knockdown.

of *MAP4K4* and *STRN4*, rescued the cells from the inhibitory effect of *shSTRN4* knockdown on AI growth (*Figure 6E*). In addition, we also observed that the expression of the dominant inhibitory K54R mutant, but not wild type MAP4K4, was able to restore the ability of these cells to form AI colonies upon *STRN4* suppression (*Figure 6—figure supplement 1D*). These observations suggest that ST inhibits MAP4K4 activity through STRN4 and the STRIPAK complex to induce transformation.

## Partial MAP4K4 knockdown induces YAP1 activation

To identify downstream signaling pathways affected during transformation by partial knockdown of *MAP4K4* expression, we performed transcriptomic profiling of HEK TER cells expressing either *shMAP4K4-82*, which induced transformation in vitro (*Figure 2B–C*), or a control shRNA targeting luciferase. We then performed a Single-sample Gene Set Enrichment Analysis (ssGSEA) (*Barbie et al., 2009*) with the *MAP4K4* knockdown gene expression signature and observed that several independent *YAP1* genesets from the literature as well as two curated *YAP1* and *TAZ* genesets from Ingenuity Pathway Analysis (IPA) were significantly associated with MAP4K4 knockdown (*Figure 7A*) (*Zhao et al., 2007*; *Yu et al., 2012*; *Hiemer et al., 2015*; *Martin et al., 2018*). We also

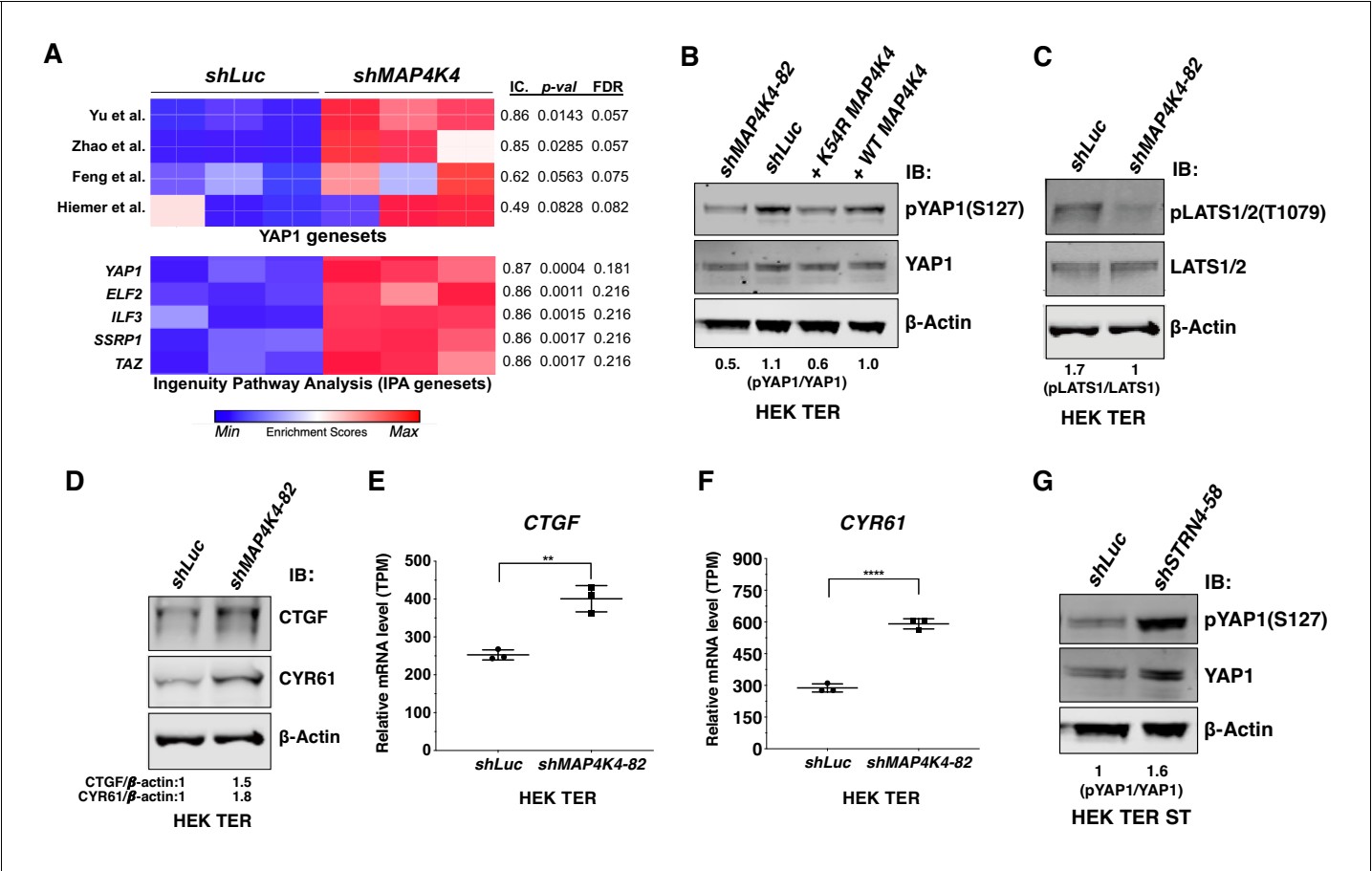

**Figure 7.** Depletion of *MAP4K4* and *STRN4* are linked to *YAP1* regulation. (**A**) Heatmap of Enrichment Scores (ES) from RNA-seq analysis showing that partial suppression of *MAP4K4* expression in HEK TER cells upregulates a transcriptional signature closely resembling four published, independently generated *YAP1* signatures and two signatures for *YAP1/TAZ* from Ingenuity Pathway Analysis (IPA) using Information Coefficient (IC) as a similarity metric. ssGSEA was performed and enrichment scores are represented as indicated in the color bar with red indicating relative enrichment and blue depletion. The three columns in the heatmap represent triplicates for each condition. (**B**) Immunoblot depicting changes in phosphorylation of YAP1 on a key negative regulatory site (S127) following partial *MAP4K4* knockdown or expression of MAP4K4 K54R in HEK TER cells. The values below the blot represent quantitation of the YAP1 pSer127 signal relative to the total YAP1 from the immunoblot. (**C**) Immunoblot showing changes in phospo-LATS1 following partial *MAP4K4* knockdown in HEK TER cells. Quantification of the LATS1 Thr1079 signal relative to total LATS1 from the immunoblot is shown below the gel. (**D**) Immunoblot depicting changes in the YAP1 target genes *CTGF* and *CYR61* following partial MAP4K4 knockdown and the ratios of the levels of CTGF/β-actin, CYR61/β-actin are shown below the blot. β-actin shown was performed in the same blot. Changes in the mRNA levels of *YAP1* target genes *CTGF* (**E**) and *CYR61* (**F**) upon *MAP4K4* suppression. (**G**) Immunoblot depicting changes in phosphorylation of YAP1 on S127 following *STRN4* knockdown in HEK TER ST. The values below the blot depict quantitation of the YAP1 pSer127 signal relative to total YAP1 from the blot (**p<0.001, ****p<0.00001).

The online version of this article includes the following source data for figure 7:

**Source data 1.** Quantification of CTGF and CYR61 gene expression (TPM).

observed that phosphorylation of YAP1 at S127, a critical, negative regulatory site that blocks nuclear import of YAP1 (*Zhao et al., 2007*), was decreased upon partial knockdown of *MAP4K4* or expression of the MAP4K4 K54R construct in HEK TER cells (*Figure 7B*). Consistent with prior reports on regulation of LATS1/2 by MAP4K4 (*Mohseni et al., 2014*; *Meng et al., 2015*; *Zheng et al., 2015*), we also found that partial knockdown of *MAP4K4* led to attenuation of p-LATS1 (*Figure 7C*). In addition, we observed that mRNA and protein levels of CTGF and CYR61, established markers of YAP1 activity, were increased upon knockdown of *MAP4K4* (*Figure 7D–F*). These observations showed that partial knockdown of *MAP4K4* at levels that induce cell transformation also led to increased YAP1 activity. In contrast, we found that suppression of *STRN4* in HEK TER

ST cells led to an increase in pYAP1 (*Figure 7G*), consistent with the change in MAP4K4 activity upon *STRN4* knockdown (*Figure 6D*).

## MAP4K4 activity converges on the regulation of the hippo/YAP1 pathway

To evaluate the role of YAP1 in transformation induced by attenuation of *MAP4K4*, we suppressed both *MAP4K4* and *YAP1* and tested AI colony formation (*Figure 8A*). We found that although knockdown of *MAP4K4* sufficed to promote transformation, we observed a three-fold decrease in AI colony growth when *MAP4K4* was co-suppressed with *YAP1* relative to an shRNA targeting luciferase as a control, indicating that transformation following partial knockdown of *MAP4K4* depends on YAP1 (*Figure 8A*, *Figure 8—figure supplement 1A*).

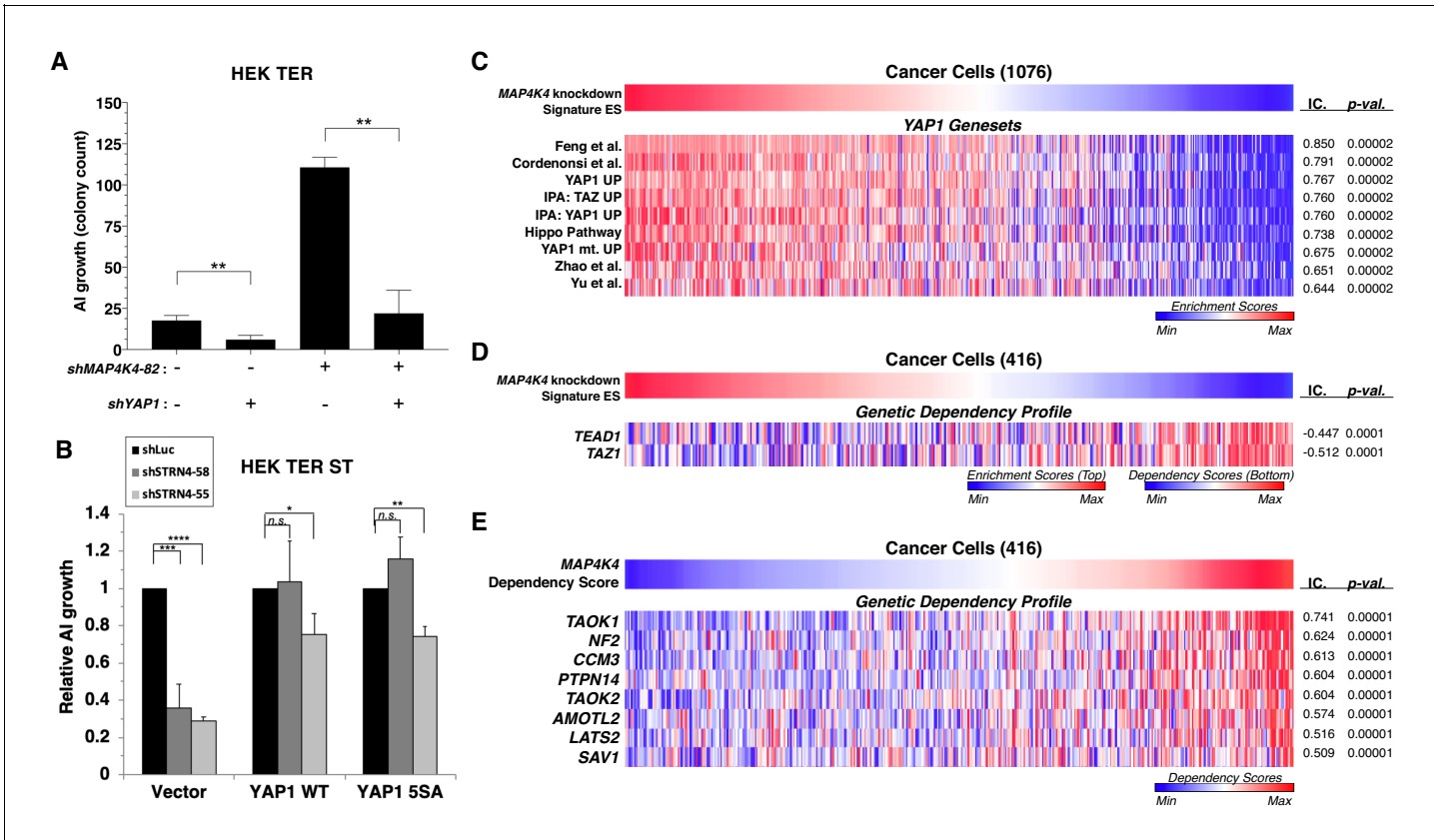

**Figure 8.** *YAP1* is necessary for transformation upon *MAP4K4* knockdown and rescues transformation in *STRN4* knockdown cells. (**A**) Quantification of AI growth obtained following partial *MAP4K4* suppression alone or when combined with *YAP1* suppression (*shYAP1*) in HEK TER cells. Transformation induced by partial *MAP4K4* suppression depends on *YAP1*. (**B**) Quantification of AI growth following *STRN4* knockdown with or without co-expression of *YAP1 WT* or the *S5A* mutant in HEK TER ST cells. YAP1 rescues the transformation defect of STRN4 suppression by shSTRN4-55 and *shSTRN4-58* (immunoblots are shown in *Figure 8—figure supplement 1B*). (**C**) Heatmap of ES depicting *YAP1* genesets from the literature significantly associated with the *MAP4K4* knockdown signature ES using Information Coefficient (IC) as a similarity metric. ssGSEA was performed using these genesets across the CCLE dataset, and enrichment scores are represented as indicated in the color bar, with red indicating relative enrichment and blue depletion (FDR < 0.0001). (**D**) Heatmap depicting top dependency genes (bottom heatmap) in the Project Achilles dependency profiles that associated with *MAP4K4* knockdown signature ES (top heatmap). The top heatmap represents ES from ssGSEA, while the bottom heatmap represent relative dependency (blue indicating strong dependency). Cell lines with low MAP4K4 transcriptional activity (in blue on top) were the most dependent on *TEAD1* and *TAZ1* (FDR < 0.0001). (**E**) Heatmap depicting co-dependency analysis of *MAP4K4* using IC across the Project Achilles data. The genes most significantly associated with *MAP4K4* dependency were enriched for the Hippo/YAP1 pathway, as well as components of the STRIPAK complex (All associations FDR < 0.0001 except *TAOK2*, *SAV1*: FDR = 0.002). (Student's t-test, *p<0.01, **p<0.001, ****p<0.00001, n.s. = not significant).

The online version of this article includes the following source data and figure supplement(s) for figure 8:

**Source data 1.** Quantification of AI growth with changes in YAP1 and MAP4K4.

**Figure supplement 1.** Changes in YAP1 and MAP4K4 protein levels and a proposed model.

To further investigate the involvement of YAP1 activity in transformation, we tested whether expression of a constitutively active YAP1 phospho-mutant allele (5SA) (*Zhao et al., 2007*) rescued transformation when *STRN4* was knocked down (*Figure 8—figure supplement 1B*). We found that reduced levels of AI growth induced by *STRN4* knockdown was rescued by expression of the wild-type or phospho-mutant *YAP1* (*Figure 8B*, *Figure 8—figure supplement 1B*). These observations show that the expression of *YAP1* or *YAP1 5SA* overrides the requirement for STRN4 in transformation.

To extend these observations beyond the HEK TER cells, we generated a *MAP4K4* knockdown gene expression signature and assessed this signature across a large collection of cancer cell lines from the Cancer Cell Line Encyclopedia (CCLE) by performing ssGSEA analysis (*Figure 8C–D*) (*Barbie et al., 2009*; *Barretina et al., 2012*). Using the resulting Enrichment Scores (ES) derived from the *MAP4K4* knockdown signature, we calculated information-theoretic measure, the Information Coefficient (IC) (*Kim et al., 2016*) to examine genesets that best matched the *MAP4K4* knockdown signature ES across these cancer cell lines. In consonance with the findings in isogenic experiments (*Figure 7A*), we observed that the *MAP4K4* knockdown signature associated significantly with a number of *YAP1* genesets derived from the literature as well as those we have generated by ectopic expression of wild-type or mutant *YAP1* in immortalized human mammary epithelial cell (YAP1 UP, YAP1 mt UP) (*Figure 8C*) (p-value<0.0001) (*Zhao et al., 2008*; *Cordenonsi et al., 2011*; *Yu et al., 2012*; *Hiemer et al., 2015*; *Feng et al., 2019*). Furthermore, when we compared the *MAP4K4* knockdown signature with gene dependency data from Project Achilles, a large-scale project involving genome-scale loss-of-function fitness screens performed in hundreds of cancer cell lines (*Aguirre et al., 2016*; *Meyers et al., 2017*; *Tsherniak et al., 2017*), we observed significant association with dependency profiles of *TAZ1* and *TEAD1* (IC = −0.447,–0.512, p-value=0.0001, 0.0001, respectively), which are both Hippo pathway effector molecules (*Figure 8D*). These findings indicated that the *MAP4K4* knockdown signature associated with dependencies in the Hippo/YAP1 pathway.

We recently showed that systematically evaluating patterns of genetic co-dependencies across a dataset identify genes with similar function (*Pan et al., 2018*). We used this same approach to examine the *MAP4K4* dependency profile. The *MAP4K4* dependency profile quantitatively reflected the relative effect of targeting *MAP4K4* on cell proliferation/survival of 416 cell lines and is represented as 'dependency scores' on cell proliferation/survival during *MAP4K4* inhibition (*Figure 8E*). To assess genes that share dependency profiles with *MAP4K4*, we performed an orthogonal analysis using IC-based associations to identify a group of genes whose dependency profiles were most significantly associated with *MAP4K4* dependency. Consistent with the known role of *MAP4K4* in regulating the Hippo pathway, we found that a number of genes whose dependency profiles were most significantly associated with those of *MAP4K4* belonged to the Hippo pathway, such as *LATS2*, *PTPN14*, and *NF2* (*Figure 8E*) (top 25 among the 18,375 dependency profiles)(p=0.00001). We also observed that *CCM3* (*PDCD10*), a member of the STRIPAK complex (*Goudreault et al., 2009*), was the top most significantly associated gene dependency with *MAP4K4*, further supporting a link between MAP4K4, Hippo, and the STRIPAK complex (*Figure 8E*). These observations suggest that the gene expression associated with *MAP4K4* knockdown is observed in many cancer cell lines and correlates with the Hippo signaling pathway.

## Discussion

Several lines of evidence now implicate the disruption of specific PP2A complexes and alteration of substrate specificity by mutation, deletion, or expression of polyomavirus ST as the basis for PP2A-mediated tumor suppressor activity. These observations have led to a model in which cancer associated PP2A mutations or ST alter the composition of PP2A complexes in cells, thus altering PP2A activity toward specific substrates. However, since purified PP2A exhibits phosphatase activity towards a broad set of substrates, the mechanisms that regulate PP2A substrate specificity in cells remains incompletely understood (*Yang et al., 1991*). Here, we show that STRIPAK regulates the interaction of PP2A with one substrate MAP4K4 that participates in PP2A-dependent cell transformation. These observations provide a mechanism by which phosphatase activity is regulated.

Previous studies had shown that most B subunits were displaced by ST from the core enzyme and could not be detected in complex with ST (*Pallas et al., 1990*; *Chen et al., 2007*; *Cho et al., 2007*;

*Sablina et al., 2010*). However, our proteomic analysis revealed that ST was bound to B‴ subunits (striatins), as well as several other STRIPAK components. Here, we evaluated whether suppressing expression of these STRIPAK components impacted ST-induced cell transformation. We found that ST expression induced increased interactions of MAP4K4 with the STRIPAK complex, which in turn reduced levels of MAP4K4 phosphorylation and activity, thus leading to increased YAP1 activity.

Prior studies have connected STRIPAK with components of the Hippo pathway (*Couzens et al., 2013*), and MAP4K4 has been shown to directly activate LATS1/2 kinases (*Meng et al., 2015*). In addition, YAP1 has been shown to be required for SV40 ST-mediated transformation (*Nguyen et al., 2014*). Recent work has shown mouse polyomavirus middle T affects YAP1 by directly binding to YAP1 and suppressing its degradation (*Hwang et al., 2014*; *Rouleau et al., 2016*). We propose a model for cell transformation induced by PP2A-mediated dephosphorylation of YAP1 in *Figure 8— figure supplement 1C*.

We have shown that partial knockdown of *MAP4K4* levels or inhibition of its kinase activity replaces ST in cell transformation, suggesting that MAP4K4 is a key PP2A substrate necessary for cell transformation. This observation is similar to our prior work that shows that only partial, but not complete, knockdown of PP2A Aα and Cα subunits leads to transformation (*Chen et al., 2004*; *Chen et al., 2005*; *Sablina et al., 2010*). However, we also note that the observed effects of suppressing *MAP4K4* leads to a greater increase in AI growth when compared to PP2A knockdown. We speculate that this may be due to the large repertoire of PP2A substrates that may have both pro-tumorigenic, as well as anti-tumorigenic activities. Likewise, MAP4K4 has been associated with a number of different pathways and biological processes (e.g., invasion, metabolism, TNF-α) and therefore, full depletion of *MAP4K4* may impact other processes that are important for transformation. Although we observed increased interactions of MAP4K4 with the STRIPAK complex in cells expressing ST, only a subset of cellular MAP4K4 interacts with STRIPAK in this context (*Figure 3C*), further supporting the notion that MAP4K4 unbound to the STRIPAK complex may have pro-tumorigenic roles. These observations also suggest that the STRIPAK complex plays a key role in regulating PP2A activity toward specific substrates and support a model in which ST in part induces transformation by promoting interactions of the STRIPAK complex with MAP4K4 and thereby attenuating MAP4K4 kinase activity, which in turn leads to the activation of YAP1.

The mechanism by which different PP2A complexes achieve substrate specificity has long remained elusive. Recent work has shown that proteins that harbor a conserved LxxIxE motif promote interactions with B56 subunits and facilitate subsequent PP2A substrate specificity (*Hertz et al., 2016*), suggesting that the substrate specificity may be achieved in part through specific interactions achieved by interactions with distinct B subunits. These findings reinforce the notion that STRIPAK serves as an organizing scaffold to bring substrates such as MAP4K4 to the PP2A complex. Indeed, recent studies have shown that MST3, a member of the STRIPAK complex, and Ste20 kinase family member MINK1 are also substrates of the STRIPAK complex (*Gordon et al., 2011*; *Hyodo et al., 2012*). It will be of interest to see if these proteins also affect transformation phenotypes in other contexts.

We found that the PP2A A-C complex continues to interact with non-canonical B‴ subunits in the presence of ST. This observation confirms prior work that showed that both STRN and STRN3 binding do not overlap with canonical B subunit binding to Aα (*Moreno et al., 2000*). Furthermore, ST has been shown to be unable to compete with and displace B subunits from interacting with the PP2A core enzyme (*Chen et al., 2007*). Indeed, early observations involving biochemical characterization of the PP2A–ST complex showed that even in the absence of canonical B subunits, PP2A bound to ST dephosphorylated histone H1, suggesting that ST may alter the substrate specificity of PP2A (*Kamibayashi et al., 1994*). Here, we provide further evidence that ST alters substrate specificity by promoting MAP4K4 interaction with the STRIPAK complex. It is unclear if ST binding to PP2A Aα promotes active conformational changes that increase PP2A A-C subunit affinity for STRN4, or if there is competition among the canonical and non-canonical B subunits to engage the PP2A core enzyme complex. However, it appears that ST interactions with STRIPAK are dependent on Aα, as ST mutants that failed to bind to Aα were also unable to bind to STRIPAK (*Figure 3—figure supplement 1A*). In addition, it was recently shown that disruption of interactions between PP2A core enzyme and canonical B subunits by mutations in PP2A Aα (P179R, R18G) promotes PP2A interactions with members of the STRIPAK complex (*Haesen et al., 2016*), reinforcing the notion that ST phenocopies the effect of cancer-associated PP2A mutations. We also observed that

the total PP2A C subunit interactions with STRN4 did not change with ST expression (*Figures 3C* and *6B–C*, *Figure 3—figure supplement 1B–C*). This finding suggests that the increased STRIPAK activity induced by ST may not be simply due to redistribution of PP2A C subunits into the STRIPAK complex but may also elicit specific changes within the STRIPAK complex. More generally, these observations suggest that striatins act as key regulators of PP2A that impart substrate specificity.

*MAP4K4* is less well characterized than other members of the MAPK family but has been implicated in a number of biological processes including invasion, insulin resistance, and immunity (*Collins et al., 2006*; *Tang et al., 2006*; *Huang et al., 2014*; *Danai et al., 2015*; *Vitorino et al., 2015*). Indeed, *MAP4K4* has been reported to promote invasion and to act as a downstream component of TNF-α signaling (*Wright et al., 2003*; *Crawford et al., 2014*; *Gao et al., 2016*). However, others have found evidence that *MAP4K4* can also act as a candidate tumor suppressor gene (*Westbrook et al., 2005*), promote apoptosis downstream of *SOX2* (*Chen et al., 2014b*; *Yang et al., 2015*) and serve as a regulator of the Hippo pathway, in part through direct phosphorylation of LATS1/2, leading to YAP/TAZ inhibition (*Couzens et al., 2013*; *Mohseni et al., 2014*; *Meng et al., 2015*; *Zheng et al., 2015*).

*YAP1* is a downstream effector of the Hippo pathway and is involved in a number of important cellular processes including organ size control and cell proliferation. When the Hippo pathway is activated by upstream stimuli triggered by cell-cell contact, cell density and detachment, YAP1 is negatively regulated through a cascade of phosphorylation events causing YAP1 to reside in the cytoplasm and remain inactive. Therefore, tight regulation of the phosphorylation and dephosphorylation events that control the Hippo pathway and subsequent YAP1 activity is critical for preserving normal cellular homeostasis. *YAP1* has also been shown to play prominent roles in oncogenic transformation, drug resistance and the epithelial-mesenchymal transition (*Hong et al., 2014*; *Shao et al., 2014*; *Wilson et al., 2015*). *YAP1* has also been shown to be required for KRAS and ST-mediated transformation, providing further evidence that *YAP1* is critical for cancer development and maintenance (*Hong et al., 2014*; *Nguyen et al., 2014*; *Shao et al., 2014*).

Despite clear evidence for *YAP1* in both cancer initiation and progression, few mutations involving *YAP1* or other Hippo pathway components have been identified in cancers. Since mutations affecting PP2A subunits are commonly observed in several types of cancer, our observation that certain PP2A complexes can activate YAP1 in the context of ST-mediated transformation suggests that these cancer-associated mutations may also serve, in part, to activate YAP1.

## Materials and methods

### Cell lines

HEK TER cells were generated from human embryonic kidney (HEK) cells, which were immortalized by introducing hTERT, SV40 Large-T antigen, and H-RAS G12V (*Hahn et al., 2002*). These cells were cultured in MEM-alpha media (Gibco) supplemented with 10% FBS. 293 T cells (ATCC) and HCT-116 (ATCC) cells were cultured in Dulbecco's modified Eagle medium (DMEM) (Cellgro) supplemented with 1% Pen-Strep (Gibco), 1% Glutamax (Gibco) and 10% fetal bovine serum (FBS) (Sigma). IMR90 cells (ATCC) were cultured in DMEM supplemented with 1% Pen-Strep, 1% Glutamax, and 1% non-essential amino acids (Gibco) and 15% FBS. For cell line identity confirmation, we utilize a Fluidigm genotyping to assay a set of 96 single nucleotide polymorphism (SNP) markers. An overlapping reference set of 42 SNPs was derived from the Affymetrix SNP6.0 array Birdseed genotype calls for cell lines also profiled in the Cancer Cell Line Encyclopedia (CCLE) project. Fingerprints (genotypes for those same SNPs) assayed by the Fluidigm assay for a particular cell line sample are compared to this reference set of SNPs across all CCLE lines, using the GenePattern FPmatching module at http://genepattern.broadinstitute.org/gp/. For cells that are not part of the CCLE project, we compare all 96 SNP markers from our Fluidigm genotyping to a collection of previously performed Fluidigm assays. In addition, we determine whether pre- and post- experimental manipulation of these samples match. In some cases, we submit cell lines for authentication by short tandem repeat (STR) profiling at DDC Medical or ATCC.

## Sample preparations for the global phosphoproteomics

HEK TER cells expressing SV40ST or suppressed the expression of PP2A Cα, Aα or B56γ subunits were synchronized in serum-free medium for 24 hr, followed by serum stimulation (5 min) and immediately harvested. Experiments were performed on two independent days as replicates.

## Global phosphoproteomics

Cell pellets were solubilized by repeated pipetting using in 10 volumes of 7.2M guanidine HCl 0.1M ammonium bicarbonate. Insoluble material was pelleted for 10 min at 10,000 x g and the protein concentration of the supernatants quantified by bicinchoninic acid assay (Pierce). Aliquots corresponding to 50 µg of each sample were transferred to new tubes and the volumes brought to 50 µl using the above solubilization buffer before further processing. Cysteine residues were reduced with 10 mM dithiothreitol (DTT) for 30 min at 56°C and alkylated with 22.5 mM iodoacetamide for 20 min at room temperature in the dark. The concentration of guanidine HCl was lowered by adding 9 volumes of 0.1M ammonium bicarbonate. Samples were digested overnight at 37°C using 10 µg of trypsin (Promega). An additional 10 µg of trypsin was added the following morning and incubated for another 4 hr at 37°C. The resulting tryptic peptide solutions were acidified by adding trifluoroacetic acid (TFA) to a final concentration of 1% and desalted on a Waters C18 solid phase extraction plate (using two consecutive passes). Eluted peptides were concentrated in a vacuum concentrator and reconstituted with 30 µL of 0.5 M triethylammonium bicarbonate. Each tube of iTRAQ reagent was reconstituted with 70 µL ethanol and added to each peptide solution. The labeling reaction was carried out for 1 hr at room temperature. Labeled peptides were combined in a tube containing 100 µL of 16.5 M acetic acid, concentrated by vacuum centrifugation and desalted on a Waters C18 solid phase extraction plate. Magnetic Fe-NTA agarose beads (300 µL of a 5% bead suspension) were prepared as described (*Ficarro et al., 2009*). The beads were added to iTRAQ labeled peptides reconstituted with 80% acetonitrile/0.1% TFA at a concentration of 0.5 µg/µL. enriched for 30 min at room temperature with end-over-end rotation. After removing the supernatant, beads were washed three times with 400 µL 80% acetonitrile/0.1% TFA, and once with 400 µL of 0.01% acetic acid. Phosphopeptides were eluted for 5 min at room temperature with 50 µL of 0.75M ammonium hydroxide containing 100 mM EDTA. The beads were washed once with 50 µL of water and this wash was combined with the eluate. Eluted phosphopeptides were concentrated to 10 µL by vacuum centrifugation. Ammonium formate (pH10) was added to yield a final concentration of 20 mM. Enriched phosphopeptides were analyzed by multidimensional RP-SAX-RP-MS/MS (*Ficarro et al., 2009*) at a depth of 43 fractions on an LTQ-velos mass spectrometer. The spectrometer was operated in data dependent mode where the top 10 most abundant ions in each MS scan were subjected to alternating CAD (electron multiplier detection, 35% normalized collision energy, q = 0.25) and HCD (image current detection, 45% normalized collision energy) MS/MS scans (isolation width = 2.0 Da (CAD) and 2.4 Da (HCD), threshold = 20,000). Dynamic exclusion was enabled with a repeat count of 1 and exclusion duration of 30 s. ESI voltage was 2.2 kV. MS spectra were recalibrated using the background ion (Si(CH3)2O)six at m/z 445.12 + /- 0.03 and converted into a Mascot generic file format (.mgf) using multiplierz scripts (*Askenazi et al., 2009*; *Parikh et al., 2009*). CAD and HCD spectra were independently searched using both Mascot (version 2.3) and Protein Pilot (version 4.5) against three appended databases consisting of: (i) human protein sequences (downloaded from RefSeq on 07/11/2011); (ii) common lab contaminants and (iv) a decoy database generated by reversing the sequences from these two databases. For Mascot searches, precursor tolerance was set to 1 Da and product ion tolerance to 0.6 Da (CAD) or 0.02 Da (HCD). We used the default settings specified for CAD and HCD spectra for Protein Pilot searches (with no precursor tolerance specified). Mascot search parameters included trypsin specificity, up to two missed cleavages, fixed carbamidomethylation (C, +57 Da) and iTRAQ8plex derivatization (K and N-terminus), variable oxidation (M, +16 Da) and phosphorylation (S, T, Y, +80 Da). Protein Pilot search parameters included trypsin specificity, fixed carbamidomethylation (C, +57 Da), peptide level iTRAQ8plex labeling. mgf files corresponding to the 43 RP-SAX-RP MS/MS fractions were individually searched with Mascot and combined into one Excel file before calculating false discovery rate (FDR). Peptide summaries were exported as text files from the Protein Pilot search results and imported into Excel for FDR calculation. Data files were processed to remove i) peptide spectral matches (PSMs) to the reverse database; PSMs to the forward database with an FDR greater than 1.0% and iii) PSMs corresponding to spectrum with no

iTRAQ reporter ions. PSMs were then compared across the four search results (Mascot CAD, Mascot HCD, Pilot CAD and Pilot HCD). PSMs with discordant peptide sequences were discarded. Peptide-level phosphorylation sites were selected based on a majority rule across searches for which an ID was made and used to locate protein-level phosphorylation in the SwissProt database (downloaded 02/06/2013; note that the entry for MAP4K4 (O95819-2) in this database contained a deletion at S627). iTRAQ intensities were summed across all PSMs with peptide sequences overlapping the protein-level phosphorylation site. The screen was performed across two replicates after randomizing the assignment of iTRAQ channel to biological samples.

## Virus production

Packaging and envelope plasmids were co-transfected with lentiviral or retroviral expression vectors into 293 T cells using Lipofectamine 2000 (Life Technologies). Two days after transfection, 293 T cell supernatant was clarified with a 0.45 µm filter and supplemented with 4 µg/mL polybrene (Santa Cruz) before transducing recipient cells. Stable cell lines were generated after selection with 2 µg/mL puromycin (Sigma), 5 µg/mL blasticidin (Invivogen), 500 µg/mL G418 (Sigma) and 50 µg/mL hygromycin (Santa Cruz) as required by each vector. For MAP4K4 inhibitor experiments, dimethyl sulfoxide (DMSO) (Sigma) or inhibitor (compound 29) (*Crawford et al., 2014*) was used at the indicated concentrations.

## Recombinant DNA constructs

MAP4K4 cDNA was generated by PCR-based Gateway cloning (Invitrogen) from HEK TER cells. NTAP-SV40 ST and NTAP-GFP have been previously described (*Rozenblatt-Rosen et al., 2012*). Mutations in MAP4K4 and SV40 ST were introduced using the QuikChange XL II site-directed mutagenesis kit (Agilent). Lentiviral shRNA constructs were obtained from the Genetic Perturbation Platform (GPP) at the Broad Institute (Cambridge, MA) (http://www.broadinstitute.org/rnai/public/). The following clone IDs were used for *STRN3*: TRCN0000365162, TRCN0000370206, *STRIP1*; TRCN0000164502, TRCN0000162951, *MARCKS*: TRCN0000197145, TRCN0000029041, and *STK24*: TRCN0000000641, TRCN0000000644. *STRN4*: TRCN0000036954 (*shSTRN4-54*), TRCN0000036955 (*shSTRN4-55*), TRCN0000036957 (shSTRN4-57), TRCN0000036958 (*shSTRN4-58*). Subsequent functional studies were carried out with TRCN0000036958 (*shSTRN4-58*) and TRCN0000036955 (*shSTRN4-55*) and as these were further confirmed to have stronger knockdown of STRN4 protein, as well as relatively less off-target effects. *STRN4* open-reading frame (ORF) construct which is resistant to *STRN4 shRNA* (TRCN0000036958 or *shSTRN4-58*) was generated by cloning of the following sequence; GCCCTTGAAGTCGAACCAATTCATGCT, which was obtained from IDT as gblocks gene fragment (Integrated DNA Technologies), into *STRN4* wild-type ORF in pdonr223 using Gibson assembly cloning kit (Cat#2611, New England Biolabs), followed by gateway cloning into pLX304 vectors from the GPP. For knockdown of *MAP4K4*, we used clone IDs: TRCN0000220092, TRCN0000220093 and TRCN0000195258, TRCN0000219681, TRCN0000219682, TRCN0000195121, TRCN0000199325. For most of the study, we focused on TRCN0000219682 (*shMAP4K4-82*) unless otherwise indicated, as described in the main text. For *YAP1 shRNA*, we used clone ID: TRCN0000107265.

For global phosphoproteomic experiments, we used PP2A shRNAs as previously described (*Sablina and Hahn, 2007*). Specifically, shRNAs targeting PP2A Cα, Aα, B56γ subunits were obtained from Genetic Perturbation Platform (GPP) with the clone IDs: TRCN0000002483 (*shPP2A Cα1*), TRCN0000002484 (*shPP2A Cα2*), TRCN0000002494 (*shPP2A B56γ1*), TRCN0000002496 (*shPP2A B56γ2*) and TRCN0000231508 (*shPP2A Aα*).

For *STRN4* CRISPR-CAS9-mediated knockout, the lentiCRISPRv2 vector was used [a gift from Feng Zhang (Addgene plasmid # 52961) (*Sanjana et al., 2014*). *STRN4*-specific sgRNA sequences were obtained from the Avana library (*Doench et al., 2016*) and sgRNAs were cloned according to Zhang lab protocols (http://genome-engineering.org/gecko/wp-content/uploads/2013/12/lenti-CRISPRv2-and-lentiGuide-oligo-cloning-protocol.pdf).

Gateway-compatible cDNA entry clones were transferred from pDONR221 or pDONR223 donor vectors to the respective retro- or lentiviral Gateway destination vectors via Gateway recombinational cloning (Life Technologies). The vectors MSCV-N-terminal-Flag-HA-IRES-PURO (NTAP) and MSCV-C-terminal-Flag-HA-IRES-PURO (CTAP), as well as all HPyV cDNAs, have been previously

described (*Sowa et al., 2009*; *Rozenblatt-Rosen et al., 2012*; *Berrios et al., 2015*). Where indicated, untagged constructs were expressed in the CTAP vector with a TAA stop codon to exclude expression of the epitope tag. Wild-type or phospho-mutant *YAP1* was cloned into pMSCV puro vector (Clontech) to generate pMSCV puro *YAP1 WT* or *5SA*.

The following plasmids were obtained from Addgene: pBabe-hygro-hTERT (plasmid # 1773) (*Counter et al., 1998*), pBabe-HcRed-Ras (plasmid # 10678) (*Boehm et al., 2005*), pBabe-neo-large T cDNA (plasmid # 1780) (*Hahn et al., 2002*), pWZL-Blast-ST (plasmid # 13805) (*Chen et al., 2004*), lentiviral packaging plasmid psPAX2 and envelope plasmid pMD2.G (plasmid #12260, #12259), retroviral packaging plasmid pUMVC3 (plasmid # 8449) (*Stewart et al., 2003*), and envelope plasmid pHCMV-AmphoEnv (plasmid # 15799) (*Sena-Esteves et al., 2004*).

## Proteomic analysis of immunopurified MAP4K4

FLAG-MAP4K4 immuno-precipitates (*Adelmant et al., 2019*) were diluted in 100 mM Ammonium Bicarbonate containing 0.1% RapiGest (final concentration) and reduced with 10 mM DTT for 30 min at 56˚C. Reduced cysteine residues were alkylated with 22.5 mM iodoacetamide for 20 min in the dark. Proteins were digested with 5 µg of trypsin overnight at 37˚C. Tryptic peptides were purified by batch-mode reverse-phase chromatography (POROS 50R2, Applied Biosystems) and subjected to immobilized metal affinity chromatography (IMAC) to enrich phosphopeptides as described for the global phosphoproteomics screen. Peptides from the IMAC supernatant were concentrated under vacuum and purified by batch-mode strong cation exchange chromatography (POROS 50HS). Phosphopeptides were analyzed by LC-MS/MS as follow: Phosphopeptides were loaded off-line onto a precolumn (4 cm POROS 10R2) and eluted with an HPLC gradient (NanoAcquity UPLC system, Waters; 5–40% B in 45 min; A = 0.2 M acetic acid in water, B = 0.2 M acetic acid in acetonitrile). Peptides were resolved on a self-packed analytical column (50 cm Monitor C18, Column Engineering) and introduced in the mass spectrometer (QExactive HF mass spectrometer, Thermo) equipped with a Digital PicoView electrospray source platform (New Objective). The mass spectrometer was operated in data-dependent mode where the top 10 most abundant ions in each MS scan were subjected to high energy collision induced dissociation (HCD, 27% normalized collision energy) and subjected to MS/MS scans (isolation width = 1.5 Da, intensity threshold = 25.000, MS1 resolution: 120 000). Dynamic exclusion was enabled with an exclusion duration of 30 s. ESI voltage was set to 3.8 kV. Dedicated MS/MS scans were also included to continuously monitor precursors for two phosphopeptides identified in a previous analysis. Peptides from the supernatant were separated using a 90 min HPLC gradient and analyzed using in the mass spectrometer as described above. MS spectra were recalibrated using the background ion (Si(CH3)2O)six at m/z 445.12 + /- 0.03 and converted into a Mascot generic file format (.mgf) using multiplierz scripts. Spectra were searched using Mascot (version 2.6) against three appended databases consisting of: i) human protein sequences (downloaded from RefSeq on 06/26/2019); ii) common lab contaminants and iii) a decoy database generated by reversing the sequences from these two databases. Precursor tolerance was set to 20 ppm and product ion tolerance was set to 25 mmu. Search parameters included trypsin specificity, up to two missed cleavages, fixed carbamidomethylation (C, +57 Da) and variable oxidation (M, +16 Da) and phosphorylation (S, T, +80 Da). Spectra matching to peptides from the reverse database were used to calculate a global false discovery rate and were discarded. The intensity of heavy and light SILAC features was directly retrieved from the mass spectrometry raw files using the multiplierz python environment (*Alexander et al., 2017*). MAP4K4 phosphorylation sites were remapped to isoform 6 (UniProt accession O95819-6) and SILAC intensities were summed for individual sites identified across overlapping peptide sequences. The SILAC intensity ratio representing the relative abundance of phosphorylation sites in cells expressing GFP or ST was normalized to correct for small difference in immunopurified MAP4K4 in the respective samples as measured in the IMAC supernatant. The relative abundance of proteins in the IMAC supernatant was calculated by summing the intensities of the heavy or light features across peptides mapping uniquely to a gene (*Askenazi et al., 2010*). Two independent FLAG-MAP4K4 immunoprecipitations were performed on combined extracts of GFP and ST expressing cells metabolically encoded with heavy and light or light and heavy SILAC labels, respectively.

## Immunoprecipitation and immunoblotting

Cell lysates were obtained using lysis buffer (150 mM NaCl, 50 mM Tris-HCl, 1 mM EDTA, 0.5% NP-40, 10% glycerol, and protease and phosphatase inhibitor cocktail sets (Calbiochem)). Immunoprecipitations were performed with protein A/G magnetic beads (Millipore) mixed with immunoprecipitation antibodies. After overnight incubation at 4°C, beads were washed with high salt lysis buffer (containing 300 mM NaCl), boiled in SDS sample buffer (Boston BioProducts), resolved by SDS-PAGE (Criterion TGX precast gels, Bio-Rad), transferred to nitrocellulose membranes (Bio-Rad), blocked and incubated with the appropriate primary antibody in TBS-T overnight at 4°C. Detection of proteins was performed with horseradish-peroxidase conjugated secondary antibodies (Rockland), developed using Clarity Western ECL substrate (Bio-Rad), and imaged with a G:BOX Chemi detection system (Syngene).

## MudPIT

HEK TER cells expressing either SV40 ST or GFP (30 × 15 cm diameter plates) were harvested with lysis buffer (20 mM imidazole HCl, 2 mM EDTA, 2 mM EGTA, pH 7.0 with 10 ug/mL each of aprotinin, leupeptin, pepstatin, 1 mM benzamidine, and 1 mM PMSF). The clarified cell extract was incubated overnight at 4C with 20–100 ug of STRN4 antibodies (Abcam, ab177155) crosslinked to 30 mg protein A agarose beads (Thermo Scientific) by dimethyl pimelimidate (DMP). Beads were washed five times with high salt lysis buffer (containing 300 mM NaCl), washed with TBS two times, and then eluted with 0.2 M glycine pH 3 and neutralized with 1 M Tris-HCl pH 8.0. Proteins were precipitated with trichloroacetic acid (20% final concentration) overnight at 4C, washed with cold acetone and processed for subsequent MudPIT analysis (*Florens and Washburn, 2006*).

In brief, TCA-precipitated protein eluates were urea-denatured, reduced, alkylated, and digested with endoproteinase LysC followed by trypsin. The peptide mixtures were loaded onto microcapillary fused silica columns (100 um i.d.), packed with C18 reverse phase (Aqua; Phenomenex), SCX (Luna; Phenomenex) and C18-RP, placed in-line with an Agilent 11000 quaternary pump, and analyzed by a 10-step MudPIT on linear ion traps. MS/MS datasets were searched using ProLuCID against a non-redundant human protein database (NCBI, 2019-12-03) containing 44,080 non-redundant human proteins, 426 usual contaminants, as well as the sequences for small and large T antigens from SV40 Macaca mulatta polyomavirus 1. To estimate false discover rates (FDRs), the amino acid sequence of each non-redundant protein was randomized (44,521 shuffled proteins) and added to the search space. Cysteine carboxylation was searched as a static modification, while methionine oxidation was searched dynamically. Peptide/spectrum matches were sorted and selected using DTASelect in combination with an in-house software, swallow, to FDRs at the peptide and protein levels of less than 1%.

## In vitro kinase assay

TAP-purified MAP4K4 eluted in standard lysis buffer with protease and phosphatase inhibitors were added to kinase assay buffer (25 mM Tris-HCl pH 7.5, 5 mM β-glycerophosphate, 2 mM dithiothreitol, 0.1 mM sodium orthovanadate and 10 mM MgCl$_2$) containing 20 μM ATPγS (Abcam) and 1 μg of myelin basic protein (MBP) (Sigma). Where specified, ATPγS was left out of the reaction as a negative control. Kinase reactions were carried out as previously described (*Allen et al., 2007*). Reactions were carried out at 30°C for 30 min. P-nitrobenzyl mesylate (PNBM) (Abcam) was then added (2.5 mM final) and the reaction was incubated at room temperature for 2 hr, followed by addition of 6x SDS loading buffer, boiling of samples, SDS-PAGE and subsequent immunoblotting for phosphorylated MBP. Relative activity was calculated as the ratio of the band intensities (measured with ImageJ) between the thiophosphate ester signal (phospho-MBP) and HA signal (NTAP-MAP4K).

## PP2A phosphatase assay

To measure PP2A phosphatase activity, we used a PP2A Immunoprecipitation Phosphatase Assay Kit (Millipore Sigma, catalog number 17–313). In brief, HEK TER GFP or ST cells were lysed in 20 mM imidazole HCl, 2 mM EDTA, 2 mM EGTA, pH 7.0 with 10 μg/mL each of aprotinin, leupeptin, pepstatin, 1 mM benzamidine, and 1 mM PMSF. Two milligrams of the lysates were then immunoprecipitated with 2 μg of anti-STRN4 antibody (Abcam, ab177155) and 40 μl of protein-A-agarose beads at 4°C overnight. Beads were washed three times with lysis buffer followed by the Ser/Thr assay buffer.

Phosphatase reactions were then performed in Ser/Thr assay buffer with a final concentration of 750 µM of MAP4K4 phosphopeptides: S771/S775 (A-A-S-pS-L-N-L-pS-N-G-E-T-E-S-V-K), S876 (L-T-A-N-E-T-Q-pS-A-S-S-T-L-Q-K) or S1251 (V-F-F-A-pS-V-R-S) for 10 min at 30°C. To provide evidence that the immunoprecipitated phosphatase activity is PP2A, we treated parallel immunoprecipitates with 5 nM of okadaic acid (Cell Signaling, #5934). Dephosphorylation of the phosphopeptide was measured through malachite green phosphate detection at 650 nm.

## AI growth and proliferation assays

HEK TER AI growth in soft agar was performed as previously described (*Hahn et al., 2002*) using 6-well dishes with BactoAgar (Gibco) at concentrations of 0.3% top and 0.6% bottom layers. Wells were fed with top agarose once per week. After 4 to 5 weeks, cells were stained with 0.005% crystal violet (Sigma) in PBS and colonies were counted. For MAP4K4 inhibitor experiments, dimethyl sulfoxide (DMSO) (Sigma) or inhibitor (compound 29) (*Crawford et al., 2014*) were used at the indicated concentrations in both the bottom and top soft agar layers and included in refeedings. For proliferation assays, cells were seeded in triplicate in 24-well plates (day 0; $5 \times 10^3$ cells per well). Cell density was measured by crystal violet assay at intervals after plating as previously described (*Rozenblatt-Rosen et al., 2012*).

## In vivo xenografts

For in vivo xenograft experiments, $2 \times 10^6$ HEK TER (expressing *shLuc* or *shMAP4K4-82*) or HEK TER ST (expressing *shLuc* or *shSTRN4-58*) cells were subcutaneously injected into the top, left and right flanks of 5 female Taconic NCR-nude (CrTac:NCr-Foxn1nu) mice. For the shSTRN4 experiments, we re-engineered the HEK TER cells to express *KRAS G12V*, because other vectors containing *HRAS G12V* with various selection markers failed to produce sufficient levels of expression. Tumor volume was assessed via caliper measurement every week by the formula: volume = length x width$^2 \times 0.5$. All procedures were performed according to protocols approved by the Institutional Animal Care and Use Committees of the Dana-Farber Cancer Institute.

## RNA-sequencing

A total of 500,000 cells of either HEK TER shLuc or *shMAP4K4-82* were seeded in three 15 cm dishes and allowed to grow for 48 hr. Total RNA was extracted using an RNeasy Plus Kit (Qiagen). RNA sequencing libraries were prepared using a NEBNext Ultra Directional RNA Library Prep Kit for Illumina, NEB E7420. The concentration of each cDNA library was quantified with the KAPA Illumina ABI Quantification Kit (Kapa Biosystems). Libraries were pooled for sequencing using the HiSeq 2500.

## Data analysis

Global phosphoproteomic data: Data from iTRAQ experiments were processed by first merging the two replicate datasets, which resulted in 6025 phosphopeptides corresponding to 2428 individual proteins. We then normalized the raw read counts of each sample to the corresponding control experiments (shRNA against luciferase for shRNA experiments, and GFP for ST experiments) followed by log$_2$ transformation. The resulting values were further normalized by quantile normalization. We performed comparative marker selection to find phosphorylation changes which are most significantly correlated with cell transformation phenotype using signal-to-ratio statistics after 1000 permutations (*Gould et al., 2006*). The transformation phenotype upon knockdown of PP2A Cα, Aα, B56γ or SV40ST expression was determined via AI growth assay described above. To facilitate direct comparison of the MAP4K4 phosphosites across different proteomic results, all MAP4K4 phosphorylation sites were mapped and compared relative to the sites in isoform 6 of the MAP4K4 protein (O95819-6) Uniprot database (https://www.uniprot.org). Raw mass spectrometry data files from SILAC and iTRAQ are available for free download at ftp://massive.ucsd.edu/MSV000084422/. MudPIT mass spectrometry data files are available for download at Massive: ftp://massive.ucsd.edu/MSV000084662/ and ProteomeXchange:http://proteomecentral.proteomexchange.org/cgi/GetDataset?ID=PXD016628.

RNAseq analysis read count was converted to Transcripts Per Million (TPM) using Kallisto quant functions (https://github.com/UCSD-CCAL/ccal) (GRCh38). Differential gene expression analysis of

samples with *MAP4K4* suppression vs. control was performed using mutual information. We also performed ssGSEA analysis of genesets from the literature, MsigDB (http://software.broadinstitute.org/gsea/msigdb/index.jsp), as well as IPA (https://www.qiagenbioinformatics.com/products/ingenuity-pathway-analysis) on the samples to obtain enrichment score for each genesets (*Zhao et al., 2007*; *Barbie et al., 2009*; *Yu et al., 2012*; *Hiemer et al., 2015*; *Martin et al., 2018*). Using the Information Coefficient (IC) (*Kim et al., 2016*), we estimated the degree of association of the phenotype (shMAP4K4 vs. shLuc) and their significance to the genesets. ssGSEA and mutual information calculations: The FDRs were computed from empirical p-values using the standard Benjamini-Hochberg procedure. The empirical p-values were obtained from an empirical permutation test where the target profile is randomly permuted to generate a null distribution for the Information Coefficient (IC) values. We also generated signatures from these experiments to apply them in the CCLE RNA Seq dataset (www.broadinstitute.org/ccle) (*Barretina et al., 2012*) using ssGSEA. Using IC, we matched top gene dependencies associated with *MAP4K4* knockdown signature score across CCLE using Gene dependency data from Project Achilles data portal using dataset version V3.12a (www.broadinstitute.org/achilles) (*Aguirre et al., 2016*; *Meyers et al., 2017*; *Tsherniak et al., 2017*).

Statistical analysis: All the student t-tests and p-value calculations were performed using GraphPad Prism software (https://www.graphpad.com). Unless indicated, experiments were performed in triplicates and the Student's t-tests were performed between perturbation and relevant control conditions using triplicates values obtained from each experiment using parametric testing. For experiments presented in *Figures 3E*, *4A*, *6D–E* and *8B*, data were first normalized to the mean of the controls and resulting mean values for each condition were plotted and error bars were calculated from standard deviation of the values.

## Acknowledgements

We thank Anna Schinzel for helping with preparing lentiviral libraries. We also thank Anna Sablina and Anne-Claude Gingras for helpful discussions. LentiCRISPRv2 was a generous gift from Feng Zhang.

## Additional information

### Competing interests

William C Hahn: Reviewing editor, *eLife*. Consultant for Thermo Fisher, AjuIB, MPM Capital, iTeos, Tyra Biosciences, Frontier Medicines and Parexel. Founder and serves on the scientific advisory board of KSQ Therapeutics. Kim Stegmaier: has previously consulted for Novartis and Rigel Pharmaceuticals and receives grant funding from Novartis on unrelated topics. Nathanael S Gray: is a founder, science advisory board member and equity holder in Gatekeeper, Syros, Petra, C4, B2S and Soltego. Also receives or has received research funding from Novartis, Takeda, Astellas, Taiho, Janssen, Kinogen, Voronoi, Her2llc, Deerfield and Sanofi. Jarrod A Marto: serves on the scientific advisory board of 908 Devices and has received sponsored research funding from AstraZeneca and Vertex. James DeCaprio: has served as a consultant to Merck & Co, Inc and has received research funding from Constellation Pharmaceuticals, Inc. The other authors declare that no competing interests exist.

### Funding

| Funder | Grant reference number | Author |
|---|---|---|
| National Cancer Institute | P01 CA203655 | James DeCaprio<br>William C Hahn |
| National Cancer Institute | U01 CA217885 | Jong Wook Kim<br>Huwate Yeerna<br>Pablo Tamayo |
| National Cancer Institute | R01 NS050674 | Guillaume Adelmant<br>Jarrod A Marto |

| National Cancer Institute | P01 CA203655 | Guillaume Adelmant<br>Jarrod A Marto |
| National Cancer Institute | R01 CA215489 | Guillaume Adelmant<br>Jarrod A Marto |

The funders had no role in study design, data collection and interpretation, or the decision to submit the work for publication.

## Author contributions

Jong Wook Kim, Conceptualization, Data curation, Formal analysis, Validation, Investigation, Visualization; Christian Berrios, Conceptualization, Data curation, Formal analysis, Investigation, Visualization, Methodology; Miju Kim, Amy E Schade, Validation, Investigation, Methodology; Guillaume Adelmant, Data curation, Formal analysis, Visualization, Methodology; Huwate Yeerna, Formal analysis, Visualization; Emily Damato, Validation; Amanda Balboni Iniguez, Resources, Investigation; Laurence Florens, Formal analysis, Methodology; Michael P Washburn, Jarrod A Marto, Supervision, Methodology; Kim Stegmaier, Resources, Supervision, Investigation; Nathanael S Gray, Resources; Pablo Tamayo, Supervision, Visualization, Methodology; Ole Gjoerup, Supervision; James DeCaprio, William C Hahn, Conceptualization, Resources, Supervision, Funding acquisition, Project administration

## Author ORCIDs

Jong Wook Kim (ID) https://orcid.org/0000-0002-3021-7193
Amy E Schade (ID) http://orcid.org/0000-0002-0342-8251
Michael P Washburn (ID) http://orcid.org/0000-0001-7568-2585
Nathanael S Gray (ID) https://orcid.org/0000-0001-5354-7403
William C Hahn (ID) https://orcid.org/0000-0003-2840-9791

## Ethics

Animal experimentation: This study was performed in strict accordance with the recommendations in the Guide for the Care and Use of Laboratory Animals of the National Institutes of Health. All of the animals were handled according to approved institutional animal care and use committee (IACUC) protocols of the Dana-Farber Cancer Institute under assurance number A3023-01. The protocol was approved by the Committee on the Ethics of Animal Experiments of the Dana-Farber Cancer Institute (Permit Number:04-101).

## Decision letter and Author response

Decision letter https://doi.org/10.7554/eLife.53003.sa1
Author response https://doi.org/10.7554/eLife.53003.sa2

# Additional files

## Supplementary files

- Supplementary file 1. Key Resources Table.

- Supplementary file 2. Normalized iTRAQ phosphoproteomic profiles of changes in phosphopeptides upon suppression of PP2A Cα, Aα, B56γ or SV40ST expression.

- Supplementary file 3. Results of the SILAC experiment representing MAP4K4 interacting proteins.

- Supplementary file 4. Results of the SILAC experiment representing targeted MAP4K4 phosphoprofiling.

- Supplementary file 5. Results of MudPIT experiment showing STRN4 interacting proteins.

- Supplementary file 6. RNAseq (TPM) profiles of MAP4K4 knockdown (shMAP4K4-82).

- Supplementary file 7. Genesets used in the study.

- Transparent reporting form

## Data availability

The RNAseq data for MAP4K4 suppression experiments have been deposited in the Gene Expression Omnibus (GEO) under accession code GSE118272. Raw mass spectrometry data files for SILAC and iTRAQ are available for free download at ftp://massive.ucsd.edu/MSV000084422/. MudPIT mass spectrometry data files are available for download at Massive: ftp://massive.ucsd.edu/MSV000084662/ and ProteomeXchange: http://proteomecentral.proteomexchange.org/cgi/GetDataset?ID=PXD016628.

The following datasets were generated:

| Author(s) | Year | Dataset title | Dataset URL | Database and Identifier |
|---|---|---|---|---|
| Kim JW, Kim M, DeCaprio J, Hahn W | 2019 | STRIPAK directs PP2A activity to promote oncogenic transformation | https://www.ncbi.nlm.nih.gov/geo/query/acc.cgi?acc=GSE118272 | NCBI Gene Expression Omnibus, GSE118272 |
| Berrios C, Florens L, Washburn MP, DeCaprio J | 2019 | MudPIT analysis of STRN4 interacting proteins from HEK TER cells expressing either SV40 ST or GFP | http://proteomecentral.proteomexchange.org/cgi/GetDataset?ID=PXD016628 | ProteomeXchange, PXD016628 |

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
