## [Decision Letter]

**Acceptance summary:**

The protein phosphatase PP2A is implicated in cancer. This is illustrated by the pro-tumorigenic effects of SV40 Small T antigen (ST) that displaces regulatory B subunits and alters the abundance and types of PP2A complexes in cells. The authors show that ST not only displaces common PP2A B subunits but also promotes A-C subunit interactions with alternative B subunits (B', striatins) that are components of the Striatin-interacting phosphatase and kinase (STRIPAK) complex. STRN4, a member of STRIPAK, associates with ST and is required for ST-PP2A-induced cell transformation. ST recruitment of STRIPAK facilitates PP2A-mediated dephosphorylation of *MAP4K4* and induces cell transformation through the activation of the Hippo pathway effector YAP1. These observations identify an unanticipated role of *MAP4K4* in transformation and show that the STRIPAK complex regulates PP2A specificity and activity.

**Decision letter after peer review:**

[Editors’ note: the authors submitted for reconsideration following the decision after peer review. What follows is the decision letter after the first round of review.]

Thank you for submitting your work entitled "STRIPAK directs PP2A activity to promote oncogenic transformation" for consideration by *eLife*. Your article has been reviewed by three peer reviewers, one of whom is a member of our Board of Reviewing Editors, and the evaluation has been overseen by a Senior Editor. The following individuals involved in review of your submission have agreed to reveal their identity: Veerle Janssens (Reviewer #2).

Our decision has been reached after consultation between the reviewers. Based on these discussions and the individual reviews below, we regret to inform you that your present manuscript will not be considered further for publication in *eLife*.

The authors have examined the mechanism of cell transformation mediated by small T (ST). The major finding reported is that ST engages the PP2A-STRIPAK complex to regulate the *MAP4K4* pathway. This is an interesting conclusion that advances the field. However, the data presented do not fully establish the PP2A-STRIPAK-ST complex and fail to document that PP2A in this complex dephosphorylates *MAP4K4*. A number of technical issues were also noted. Moreover, insufficient information was presented concerning the detailed description of the reported studies. The reviewers considered that the required revisions to the manuscript would take more than 2 months. Consequently, we are returning the manuscript to you. However, we remain interested in the mechanism of ST function within the STRIPAK complex and would be willing to consider a new manuscript that fully addresses the points raised during review.

Reviewer #1:

This is an interesting study of cellular transformation caused by the viral oncoprotein small T (ST). Previous studies have established a role for PP2A, an enzyme that binds ST. Furthermore, the authors have previously reported that ST interacts with the STRIPAK complex plus *MAP4K4*. Moreover, it is established that *MAP4K4* acts an inhibitor of the hippo pathway and it is known that hippo signaling is required for ST-mediated cellular transformation. Together, these previous analyses provide an outline for a potential ST-mediated oncogenic pathway involving PP2A, STRIPAK, *MAP4K4*, and hippo pathway signaling. The purpose of the current study was to obtain molecular insight into the functioning of this pathway.

Specific Comments

1) The study is very poorly presented. For example, the first three lines of the Results section refer to Table S1 and Figure 1—figure supplement 1A with little experimental description. This minimalist approach to description is widespread throughout the figure legends. For example, the shRNA nomenclature (e.g name and color Figure 1—figure supplement 1B) is not explained and the choice of panels is unclear – why is MARCKs presented, but not PP2A, B56, Aα? The STRN3, MAP2K3 and STK24 shRNAs are not described (Figure S4). Of the 8 *MAP4K4* shRNAs (Figure 1D), described, only six are described in the Materials and methods. The STRN4 shRNA in Figure 4D is not described.

2) Some of the conclusions presented are not supported by the presented Figure panels. For example, the "significantly inhibited as measured by in vitro kinase assays" in paragraph two of the Results is not supported by statistical analysis in corresponding Figure 1—figure supplement 1C. Similarly, the conclusion that the exon 22 encoded phosphorylation sites are regulatory based on the exon 22 deletion is not rigorous (subsection “Consequences of STRIPAK-MAP4K4 interactions on MAP4K4 phosphorylation and activity”).

3) The variability of the shRNA data is confusing. What are we to make of the STRN3 shRNAs in Figure S4? In the case of *MAP4k4*, the variability is proposed to be mediated by the extent of *MAP4K4* knockdown – this conclusion is based on the correlation between a 50% knockout down phenotype and the lack of phenotype caused by more extensive knockdown. Can this conclusion be corroborated by an experimental test – for example, CRISPR-mediated clines with heterozygous vs homozygous gene deletions?

4) One conclusion of the study is that while ST is thought to inhibit PP2A, here the authors indicate that ST bound PP2A in the STRIPAK complex is active and that it dephosphorylates *MAP4K4*. This is an interesting conclusion that advances the field. However, data are needed to confirm this conclusion. Can PP2A activity be detected in the ST/STRIPAK complex?

5) The organization of the ST/STRIPAK complex is unclear from the data presented. Is the ST association with STRIPAK mediated by the interaction of ST with PP2A? Are there ST interactions with other STRIPAK components?

6) The sole reliance on agar assays for measurements of oncogenesis needs to be justified.

Reviewer #2:

This is an interesting study describing a novel mechanism by which small t achieves cell transformation through an effect on the B"'/striatin-PP2A complexes, which on their turn suppress *MAP4K4* activity, presumably by direct dephosphorylation. Suppressed *MAP4K4* activity then further contributes to suppressed Hippo signaling, resulting in activation of the YAP1 oncogene, and increased transcription of YAP1 target genes.

The authors show that cell transformation by ST is (in part) dependent on STRN4 (or STRN3), and that ST increases STRN4 interaction with PP2A-C, and STRN4 interaction with *MAP4K4*. Although all striatins (STRN, STRN3 and STRN4) are identified as part of the ST interactome (Table S1), it is not clear from the current experiments though whether a ST-STRN4-PP2A A-C complex is indeed formed (e.g. PP2AC is missing in Table S1; and ST IB is missing in STRN4 IP in Figure 2C), and more importantly, whether such a complex is catalytically active. This is critical, not only to sustain their model (Figure 6H), but also because the prevailing idea, supported by crystallographic studies, is that ST, through binding to the A subunit of PP2A, inhibits activity of the PP2A C subunit. Thus, authors should directly address this, and isolate STRN4 from cells with or without ST expression, and subsequently measure and compare STRN4-associated PP2A activity in these two conditions. They should also check for the presence of ST in these STRN4 IPs. The activity assay could be executed on a synthetic phospho-peptide deriving from one of the *MAP4K4* phosphosites that are presumably being dephosphorylated by STRN4-PP2A, or on a more general phosphopeptide. This is for me a key experiment that is missing: only if clear PP2A activities can be measured in STRN4 IPs from cells expressing ST, and ST can be shown to be present in these STRN4 IPs, the major conclusions of the paper and the proposed model (Figure 6H) are supported. If not, the effect of ST may indeed rather be explained by promoting the formation of Striatin-PP2A complexes at the expense of other PP2A trimers, thus, by shifting the stoichiometric balance between the different PP2A holoenzymes (a possibility also suggested by the authors themselves in the Discussion).

This being said, the authors further show via a co-depletion experiment that the role of STRN4 in promoting ST-mediated transformation requires *MAP4K4*, and more specifically, is dependent on suppression of MAP4K kinase activity. *MAP4K4* isolated from ST-expressing cells was indeed found to be less active than in cells without ST expression, and this correlated with *MAP4K4* dephosphorylation at several Ser sites. *MAP4K4* dephosphorylation of these same sites was also seen in cells with knockdown of PP2A A subunit, suggesting that PP2A is directly (or indirectly) involved. In concordance, (partial) suppression of MAP4K expression by itself, is as strongly promoting cell transformation as expression of ST, and this effect is again attributable to a (partial) suppression of *MAP4K4* kinase activity, further confirming that MAP4K is important to prevent cell transformation, and thus, in principle is an additional, appealing cellular target for ST.

While suppression of *MAP4K4* does not affect the interactions of STRIPAK components (including STRN4) with PP2A (Figure 5—figure supplement 1A), the authors argue that suppression of STRN4 affects the interaction of *MAP4K4* with PP2A in ST expressing cells (Figure 5C). In my opinion, this is not very well sustained by the data, as an IP of *MAP4K4* in conditions with or without *shSTRN4* is clearly lacking in this figure, and should be added. In any case, this figure also needs quantification of the bands (and calculation of ratios) to sustain any conclusions.

Overall, the manuscript lacks at least two key experiments to fully support the model and some of the conclusions drawn (i.e. PP2A-STRN4 activity assay in cells with and without ST; and PP2A C interaction assay in *MAP4K4* IP in cells with and without *shSTRN4*).

Reviewer #3:

The submitted paper 'STRIPAK directs PP2A activity to promote oncogenic transformation', by Kim et al., demonstrate that SV40 small t antigen (ST) induce *MAP4K4* binding to STRN4 in the STRIPAK complex. The authors suggest that this interaction lowers *MAP4K4* activity and that this drives anchorage-independent (AI) cell growth due to increased YAP-signaling in these HEK293 cells. The connection between *MAP4K4* and Hippo/YAP signaling is not novel. The connection between *MAP4K4* and STRIPAK is not novel either. But, the finding that SV can induce transformation through the STRIPAK/*MAP4K4* and through changes in the PP2A complex interactome is novel.

The main conclusion is based on the fact that ST induce a 2-3 fold binding of MAP4K to the STRIPAK complex and that this decrease *MAP4K4* phosphorylation and activity. This is a novel finding, but in reality, how much of MAP4K is actually bound to the STRIPAK complex vs. non-STRIPAK complex? It would have been nice to validate these data in some other cell lines. The data in HEK293 cells stand a bit alone and never becomes really powerful. The authors also do not demonstrate if the PP2A per se is responsible for the lower MPA4K4 phosphorylation.

The paper is interesting but the data has a high degree of inconsistency, in particular when correlating the AI growth with the protein knockdown efficiencies using different ablation techniques. In many experiment there is no consistency between protein levels using different shRNA, gRNAs and the AI growth (Figure 1B, 4B and E). They explain these discrepancies by arguing that intermediate levels of *MAP4K4* are need to induce transformation – not a complete suppression!! This makes the data hard to interpret and hard to accept. The explanation is plausible and is supported by the *MAP4K4* inhibitor experiment. However, more emphasis should be made on this intermediated level of *MAP4K4* levels to convince the reader. I.e. it would be nice to see if higher conc. of Compound 29 (3-5 μM) completely inhibit AI growth. Again more cell lines should be validated.

Could the data be explained by redundancy from MAP4K6 (MINK1) and MAP4K7 (TNIK) which both show redundancy with *MAP4K4* in activity and binding to the STRIPAK complex as demonstrated in the papers:

-Misshapen-like kinase 1 (MINK1) Is a Novel Component of Striatin-interacting Phosphatase and Kinase (STRIPAK) and Is Required for the Completion of Cytokinesis and The Ste20 Family Kinases *MAP4K4*, MINK1

-TNIK Converge to Regulate Stress-Induced JNK Signaling in Neurons

The text is poorly written and the figure legends are not helping the reader. For instance, when comparing shLuc in Figure 1, the AI is around 20, when looking at shLuc in Figure 3 the AI is around 100-150. In this case, the authors should state in the figures and legends if the cells used in the experiment were transformed or not prior to performing the experiment.

Most confusing is when the authors keep talking about a *MAP4K4* signature, which in reality is a 'loss-of-*MAP4K4*' signature in the entire Figure 6. The authors need to run through their text and fix all these issues. I.e they write 'indicating that YAP1 is necessary for *MAP4K4* mediated transformation. This statement is false; it should be opposite: indicating that YAP1 is necessary for loss-of-*MAP4K4* mediated transformation.

The reader is kept with certain doubt all the time, when reading the paper and one never really gets convinced by the end. This is a pity! A massive editing (text, figure and legends) is needed to give this a nice flow. Additional validation of shRNA constructs and additional cell lines should be a requisite before acceptance.

[Editors’ note: further revisions were suggested prior to acceptance, as described below.]

Thank you for submitting your article "STRIPAK directs PP2A activity toward *MAP4K4* to promote oncogenic transformation" for consideration by *eLife*. Your article has been reviewed by two peer reviewers, one of whom is a member of our Board of Reviewing Editors, and the evaluation has been overseen by Anna Akhmanova as the Senior Editor. The reviewers have opted to remain anonymous.

The reviewers have discussed the reviews with one another and the Reviewing Editor has drafted this decision to help you prepare a revised submission.

Summary:

The authors have significantly improved the presentation of this study of PP2A regulation of *MAP4K4* mediated by STRIPAK. Moreover, the manuscript presents additional experiments to support the conclusions presented, including a phosphoproteomics experiment in which decreased phosphorylation of *MAP4K4* was observed in cells with partial PP2A suppression (via C or B56gamma-directed shRNA) and in cells expressing ST, and additional in vivo xenograft assays to further sustain the conclusions from the in vitro colony growth assays.

While the manuscript has been improved, some questions raised during the initial review have not been fully addressed and some questions have been raised by the new data that have now included.

Essential revisions:

1) Figure 1 and Figure 1—figure supplement 1: The new phosphoproteomics experiment introducing the manuscript is confusing in the way it is presented. Different results are obtained for shRNA to the same target. Is this due to differences in the extent of knockdown? A western blot to document expression levels would be helpful. Would it be simpler to present data from those shRNA that do cause transformation compared to the shLuc control? Data from the GFP control (panel 1A) are not presented. The heat map scale (panel 1B) should provide a numbers.

2) A key conclusion of the previous version of this manuscript was *MAP4K4*-directed PP2A-STRN activity in the presence of ST. A direct experimental validation of this conclusion was requested. The revised manuscript includes this experiment, but as stated by the authors, the presence of ST in the PP2A complex could not be confirmed because of reagent issues (ST co-migrates with small IgG band). It is unclear how the data presented were normalized – to STRN4 inputs, or to co-IP-ed PP2A-C levels in the STRN4 IPs? The authors seem to have used equal amounts of total lysate in both conditions to IP STRN4 from, but this does not suffice to show that equal STRN4 levels were IP-ed, let alone that equal PP2A-C levels were present in both IPs. Could the ST levels in these IPs be determined by an MS-based approach? Could recombinant ST be added to STRN4 IPs and the effect on PP2A activity measured?

3) In Figure 6B, more PP2A-C is co-IPed with STRN4 in ST overexpressing cells, while in Figure 3—figure supplement 1B this does not seem to be the case (in Figure 3C it is more difficult to judge). If more PP2A-C is co-IPed with STRN4 in ST versus GFP overexpressing cells, this would provide evidence for a relative redistribution of PP2A-C subunits into STRN4 complexes versus other PP2A B subunit-type complexes upon ST expression, and provide an explanation for the observed increase in activity. If, however, this would not be the case, and the presence of ST can be demonstrated in the complex, this would suggest that ST might act as 'an activator' of STRN4-PP2A-mediated dephosphorylation of certain *MAP4K4* sites. These questions lead to a lack of clarity concerning the mechanism-of-action of how ST stimulates STRN4-PP2A activity towards *MAP4K4*.

4) Figure 6—figure supplement 1C: indications of statistical significance are missing

---

## [Author Response]

[Editors’ note: the authors resubmitted a revised version of the paper for consideration. What follows is the authors’ response to the first round of review.]

Reviewer #1:This is an interesting study of cellular transformation caused by the viral oncoprotein small T (ST). Previous studies have established a role for PP2A, an enzyme that binds ST. Furthermore, the authors have previously reported that ST interacts with the STRIPAK complex plus MAP4K4. Moreover, it is established that MAP4K4 acts an inhibitor of the hippo pathway and it is known that hippo signaling is required for ST-mediated cellular transformation. Together, these previous analyses provide an outline for a potential ST-mediated oncogenic pathway involving PP2A, STRIPAK, MAP4K4, and hippo pathway signaling. The purpose of the current study was to obtain molecular insight into the functioning of this pathway.Specific Comments1) The study is very poorly presented. For example, the first three lines of the Results section refer to Table S1 and Figure 1—figure supplement 1A with little experimental description. This minimalist approach to description is widespread throughout the figure legends. For example, the shRNA nomenclature (e.g name and color Figure 1—figure supplement 1B) is not explained and the choice of panels is unclear – why is MARCKs presented, but not PP2A, B56, Aα? The STRN3, MAP2K3 and STK24 shRNAs are not described (Figure S4). Of the 8 MAP4K4 shRNAs (Figure 1D), described, only six are described in the Materials and methods. The STRN4 shRNA in Figure 4D is not described.

We agree with the reviewer that the original manuscript did not fully explain the figures and that the presentation of the manuscript could have been clearer. In this revised version, we have substantially revised the manuscript, figures and legends to address these concerns. First, we have completely reorganized the manuscript and now present the phosphoproteomics analysis that led to the finding that *MAP4K4* phosphorylation changes when SV40 ST is present. Second, we added substantially more detail to the description of the results throughout the manuscript and have rewritten the figure legends. Third, we also added additional descriptions regarding the choice of specific shRNAs, their designations, and additional details in the Materials and methods section of the manuscript. We have also carefully edited the manuscript to add information that was omitted in the original manuscript. We thank the reviewers for their forbearance and agree that changes have substantially improved the manuscript.

2) Some of the conclusions presented are not supported by the presented Figure panels. For example, the "significantly inhibited as measured by in vitro kinase assays" in paragraph two of the Results is not supported by statistical analysis in corresponding Figure 1—figure supplement 1C. Similarly, the conclusion that the exon 22 encoded phosphorylation sites are regulatory based on the exon 22 deletion is not rigorous (subsection “Consequences of STRIPAK-MAP4K4 interactions on MAP4K4 phosphorylation and activity”).

We agree with the reviewer. We have confirmed that when significant is used, that the corresponding experiments are accompanied by statistical analyses. We have removed the section on exon 22 entirely, as it does not contribute towards the overall conclusion of the manuscript.

3) The variability of the shRNA data is confusing. What are we to make of the STRN3 shRNAs in Figure S4? In the case of MAP4k4, the variability is proposed to be mediated by the extent of MAP4K4 knockdown – this conclusion is based on the correlation between a 50% knockout down phenotype and the lack of phenotype caused by more extensive knockdown. Can this conclusion be corroborated by an experimental test – for example, CRISPR-mediated clines with heterozygous vs homozygous gene deletions?

The reviewer is correct that different RNAi reagents induce variable levels of mRNA suppression. For *MAP4K4*, we tested 8 distinct shRNAs targeting different sequences and found that the three shRNAs that induced anchorage independent growth induced partial knockdown of *MAP4K4* (Figure 2—figure supplement 1B). To provide stronger evidence to support these experiments, we have added mRNA expression data that confirms that *MAP4K4* mRNA is reduced by 50% upon expressing the *MAP4K4* shRNA (Figure 2—figure supplement 1C). In addition, we have now performed xenograft experiments in which HEK TER cells harboring partial *MAP4K4* suppression induce tumors in immunodeficient mice when xenografted (Figure 2D-E).

The reviewer suggested an experiment involving CRISPR-Cas9 to create heterozygous vs. homozygous knockout cells. This experiment will require single cell cloning and the testing of hundreds of clones that harbor the correct genotype to ensure that any observation is not the result of selection of rare clones.

In this case, we simply wish to make the argument that partial suppression of *MAP4K4* leads to cell transformation. We include the new data described above and have clarified the text to make this point clearer.

In addition, since the STRN3 shRNA data does not contribute to the study in a meaningful way, we removed the STRN3 result originally in Figure S4 to make the manuscript more focused.

4) One conclusion of the study is that while ST is thought to inhibit PP2A, here the authors indicate that ST bound PP2A in the STRIPAK complex is active and that it dephosphorylates MAP4K4. This is an interesting conclusion that advances the field. However, data are needed to confirm this conclusion. Can PP2A activity be detected in the ST/STRIPAK complex?

We have performed additional experiments to address this important point. We isolated STRN4 immune complexes and tested PP2A phosphatase activity using *MAP4K4* phosphopeptides identified in the phosphoproteomic experiments. Specifically, we found that PP2A activity was >2 fold elevated in ST expressing cells compared to the control (Figure 3E). Since treatment with okadaic acid abolishes this activity whether or not ST is expressed, this activity is attributable to PP2A. We thank the reviewer for suggesting this experiment.

5) The organization of the ST/STRIPAK complex is unclear from the data presented. Is the ST association with STRIPAK mediated by the interaction of ST with PP2A? Are there ST interactions with other STRIPAK components?

To further demonstrate that ST interaction with STRIPAK is mediated by PP2A, we have performed additional experiments in which we expressed and tested ST mutants that are unable to bind to PP2A Aα. We found that these ST mutants are also unable to bind to STRN3, a core component of the STRIPAK (new Figure 3—figure supplement 1A). We previously showed that ST interacts with STRN3 and other STRIPAK components including (STRN4, CTTNBP2NL, FAM40A, MAP2K3, STK24) (Rozenblatt-Rosen et al., 2012).

6) The sole reliance on agar assays for measurements of oncogenesis needs to be justified.

We agree with the reviewer that additional transformation assays would enhance the manuscript. We have now incorporated in vivo tumor xenograft experiments to further support our conclusions. We demonstrate in Figure 2D-E that partial knockdown of *MAP4K4* in HEK TER cells induces tumor growth in immunodeficient mice and furthermore, as shown in Figure 5E-F, suppression of STRN4 in HEK TER ST cells substantially reduces tumor growth.

Reviewer #2:This is an interesting study describing a novel mechanism by which small t achieves cell transformation through an effect on the B"'/striatin-PP2A complexes, which on their turn suppress MAP4K4 activity, presumably by direct dephosphorylation. Suppressed MAP4K4 activity then further contributes to suppressed Hippo signaling, resulting in activation of the YAP1 oncogene, and increased transcription of YAP1 target genes.The authors show that cell transformation by ST is (in part) dependent on STRN4 (or STRN3), and that ST increases STRN4 interaction with PP2A-C, and STRN4 interaction with MAP4K4. Although all striatins (STRN, STRN3 and STRN4) are identified as part of the ST interactome (Table S1), it is not clear from the current experiments though whether a ST-STRN4-PP2A A-C complex is indeed formed (e.g. PP2AC is missing in Table S1; and ST IB is missing in STRN4 IP in Figure 2C), and more importantly, whether such a complex is catalytically active. This is critical, not only to sustain their model (Figure 6H), but also because the prevailing idea, supported by crystallographic studies, is that ST, through binding to the A subunit of PP2A, inhibits activity of the PP2A C subunit. Thus, authors should directly address this, and isolate STRN4 from cells with or without ST expression, and subsequently measure and compare STRN4-associated PP2A activity in these two conditions. They should also check for the presence of ST in these STRN4 IPs. The activity assay could be executed on a synthetic phospho-peptide deriving from one of the MAP4K4 phosphosites that are presumably being dephosphorylated by STRN4-PP2A, or on a more general phosphopeptide. This is for me a key experiment that is missing: only if clear PP2A activities can be measured in STRN4 IPs from cells expressing ST, and ST can be shown to be present in these STRN4 IPs, the major conclusions of the paper and the proposed model (Figure 6H) are supported. If not, the effect of ST may indeed rather be explained by promoting the formation of Striatin-PP2A complexes at the expense of other PP2A trimers, thus, by shifting the stoichiometric balance between the different PP2A holoenzymes (a possibility also suggested by the authors themselves in the Discussion).

We have now performed new experiments as suggested by the reviewer. Consistent with the overarching model, we have added an experiment which shows that in ST-expressing cells, STRN4 immune complexes contain >2 fold higher activity for 2 specific *MAP4K4* phosphopeptides that were identified in our phosphoproteomic analysis (Figure 3D-E) and that this activity is abolished in the presence of okadaic acid, a potent inhibitor of PP2A. Although we detected an increase in PP2A phosphatase activity in vitro using STRN4 immune complexes derived from ST expressing cells relative to GFP control, we were unable to detect ST in these complexes due to technical reasons. Specifically, the ST specific antibody cannot be used in a co-immunoprecipitation experiment because SV40 ST (21kDa) overlaps with light chain of IgG (25kDa).

As mentioned above, we found an increase in PP2A activity for specific *MAP4K4* phosphosites when ST is expressed. This observation is consistent with our results showing an increased association between striatins and PP2A (Figure 6B) and between STRIPAK and *MAP4K4* (Figure 3B-C, Figure 6B). However, we cannot exclude the possibility that ST displaces canonical B subunits and is consequently shifting the balance towards striatin-containing PP2A holoenzymes. We describe these points in the Discussion section of the manuscript.

This being said, the authors further show via a co-depletion experiment that the role of STRN4 in promoting ST-mediated transformation requires MAP4K4, and more specifically, is dependent on suppression of MAP4K kinase activity. MAP4K4 isolated from ST-expressing cells was indeed found to be less active than in cells without ST expression, and this correlated with MAP4K4 dephosphorylation at several Ser sites. MAP4K4 dephosphorylation of these same sites was also seen in cells with knockdown of PP2A A subunit, suggesting that PP2A is directly (or indirectly) involved. In concordance, (partial) suppression of MAP4K expression by itself, is as strongly promoting cell transformation as expression of ST, and this effect is again attributable to a (partial) suppression of MAP4K4 kinase activity, further confirming that MAP4K is important to prevent cell transformation, and thus, in principle is an additional, appealing cellular target for ST.While suppression of MAP4K4 does not affect the interactions of STRIPAK components (including STRN4) with PP2A (Figure 5—figure supplement 1A), the authors argue that suppression of STRN4 affects the interaction of MAP4K4 with PP2A in ST expressing cells (Figure 5C). In my opinion, this is not very well sustained by the data, as an IP of MAP4K4 in conditions with or without shSTRN4 is clearly lacking in this figure, and should be added. In any case, this figure also needs quantification of the bands (and calculation of ratios) to sustain any conclusions.

In the revised manuscript, we have added Figure 6—figure supplement 1A, where we demonstrate that knockdown of STRN4 also reduces interaction of *MAP4K4* with PP2A Cα in ST-expressing cells. As the reviewer suggested, we quantified the blots and added the ratio calculations into Figure 6—figure supplement 1A.

Overall, the manuscript lacks at least two key experiments to fully support the model and some of the conclusions drawn (i.e. PP2A-STRN4 activity assay in cells with and without ST; and PP2A C interaction assay in MAP4K4 IP in cells with and without shSTRN4).

As described above, we have now performed both of these experiments (Figure 3E and Figure 6—figure supplement 1A).

Reviewer #3:The submitted paper 'STRIPAK directs PP2A activity to promote oncogenic transformation', by Kim et al., demonstrate that SV40 small t antigen (ST) induce MAP4K4 binding to STRN4 in the STRIPAK complex. The authors suggest that this interaction lowers MAP4K4 activity and that this drives anchorage-independent (AI) cell growth due to increased YAP-signaling in these HEK293 cells. The connection between MAP4K4 and Hippo/YAP signaling is not novel. The connection between MAP4K4 and STRIPAK is not novel either. But, the finding that SV can induce transformation through the STRIPAK/MAP4K4 and through changes in the PP2A complex interactome is novel.The main conclusion is based on the fact that ST induce a 2-3 fold binding of MAP4K to the STRIPAK complex and that this decrease MAP4K4 phosphorylation and activity. This is a novel finding, but in reality, how much of MAP4K is actually bound to the STRIPAK complex vs. non-STRIPAK complex? It would have been nice to validate these data in some other cell lines. The data in HEK293 cells stand a bit alone and never becomes really powerful.

The reviewer is correct in pointing out that only a fraction of *MAP4K4* is bound to STRIPAK as demonstrated in the Co-IP study shown in Figure 3C. This is not surprising since phosphatases do not form tight complexes with substrates, and we envision that STRIPAK allows *MAP4K4* to be modified by PP2A where it then dissociates from STRIPAK. To further validate these observations in other cell contexts, in addition to HEK TER cells which are Immortalized epithelial cells from kidney and distinct from HEK293 cells, we have used IMR90 as well as HEK293T for some of the studies, thus confirming the interactions in other cell types (Figure 3—figure supplement 1A-B). We also applied the *MAP4K4* knockdown signature across 416 cell lines to apply some of our results in a broader context (Figure 8C-D).

The authors also do not demonstrate if the PP2A per se is responsible for the lower MPA4K4 phosphorylation.

Please also see the response to reviewer 1, point 4. Our new data (Figure 3E) demonstrates that PP2A activity in ST-expressing cells relative to GFP control is elevated by >2 fold towards 2 independent phosphopeptides derived from *MAP4K4*. To confirm that the dephosphorylation occurs through PP2A, we showed that it can be blocked by the specific inhibitor okadaic acid.

The paper is interesting but the data has a high degree of inconsistency, in particular when correlating the AI growth with the protein knockdown efficiencies using different ablation techniques. In many experiment there is no consistency between protein levels using different shRNA, gRNAs and the AI growth (Figure 1B, 4B and E).

We have quantified the immunoblots to clarify the relationship between protein levels induced by the shRNAs and sgRNAs and transformation phenotypes (AI and tumor growth) (Figure 5—figure supplement 1B, D). In addition, the reviewer highlights a well-known issue with AI assays. The background of such assays varies sometimes widely between experiments, and we have focused on the differences between control and experimental groups rather than absolute values. To further support these observations, we have now included xenograft experiments that confirm the findings in the AI assays.

They explain these discrepancies by arguing that intermediate levels of MAP4K4 are need to induce transformation – not a complete suppression!! This makes the data hard to interpret and hard to accept. The explanation is plausible and is supported by the MAP4K4 inhibitor experiment. However, more emphasis should be made on this intermediated level of MAP4K4 levels to convince the reader. I.e. it would be nice to see if higher conc. of Compound 29 (3-5 μM) completely inhibit AI growth. Again more cell lines should be validated.

There are many genes that when fully depleted are required for cell proliferation or survival but exhibit different phenotypes when partially suppressed or inhibited. We have previously shown that the core subunits of PP2A exhibit this behavior (PP2A A and C subunits). Indeed, such hypomorphic phenotypes are likely induced by most therapeutic agents that do not completely inhibit their targets in vivo. We have added additional text in the Discussion to clarify this phenotype of *MAP4K4*, which we have also observed with PP2A Aα and Cα. Specifically, germline deletion of *MAP4K4* is embryonic lethal. Yet, we, as well as others (Westbrook et al., 2005), have found that partial knockdown of *MAP4K4* can drive oncogenic transformation.

Could the data be explained by redundancy from MAP4K6 (MINK1) and MAP4K7 (TNIK) which both show redundancy with MAP4K4 in activity and binding to the STRIPAK complex as demonstrated in the papers:-Misshapen-like kinase 1 (MINK1) Is a Novel Component of Striatin-interacting Phosphatase and Kinase (STRIPAK) and Is Required for the Completion of Cytokinesis and The Ste20 Family Kinases MAP4K4, MINK1-TNIK Converge to Regulate Stress-Induced JNK Signaling in Neurons

We have evaluated whether MINK1 interacts with STRIPAK and have not found an interaction in this context. Furthermore, we did not recover any of these kinases in the ST proteomic analysis (Rozenblatt- Rosen et al., 2012). We have added this point to the Discussion section of the manuscript.

The text is poorly written and the figure legends are not helping the reader. For instance, when comparing shLuc in Figure 1, the AI is around 20, when looking at shLuc in Figure 3 the AI is around 100-150. In this case, the authors should state in the figures and legends if the cells used in the experiment were transformed or not prior to performing the experiment.

The reviewer is correct, and we apologize for the lack of clarity and completeness in the original submission. We have completely revised the text and the figure legends in this revised version.

In addition, the reviewer highlights a well-known issue with AI assays. The background of such assays varies sometimes widely between experiments, and we have focused on the differences between control and experimental groups rather than absolute values. To further support these observations, we have now included xenograft experiments that confirm the findings in the AI assays.

Most confusing is when the authors keep talking about a MAP4K4 signature, which in reality is a 'loss-of-MAP4K4' signature in the entire Figure 6.

We have revised the text to indicate that this is a *MAP4K4* knockdown signature. We thank the reviewer for pointing this out.

The authors need to run through their text and fix all these issues. I.e they write 'indicating that YAP1 is necessary for MAP4K4 mediated transformation. This statement is false; it should be opposite: indicating that YAP1 is necessary for loss-of-MAP4K4 mediated transformation.

We have corrected this mistake. More generally, we have completely revised the manuscript to address these issues in the original submission.

The reader is kept with certain doubt all the time, when reading the paper and one never really gets convinced by the end. This is a pity! A massive editing (text, figure and legends) is needed to give this a nice flow. Additional validation of shRNA constructs and additional cell lines should be a requisite before acceptance.

We agree and have completely revised the manuscript and figure legends. We have re-organized the presentation of the manuscript to make the points more clearly. In addition, we have performed a substantial number of new experiments. We believe that these changes have strengthened the manuscript considerably and thank the reviewers for all of their thoughtful and constructive criticisms.

[Editors’ note: what follows is the authors’ response to the second round of review.]

Essential revisions:1) Figure 1 and Figure 1—figure supplement 1: The new phosphoproteomics experiment introducing the manuscript is confusing in the way it is presented. Different results are obtained for shRNA to the same target. Is this due to differences in the extent of knockdown? A western blot to document expression levels would be helpful. Would it be simpler to present data from those shRNA that do cause transformation compared to the shLuc control? Data from the GFP control (panel 1A) are not presented. The heat map scale (panel 1B) should provide a numbers.

We and others (Sablina et al., 2010) have shown that complete suppression of PP2A Ca induces cell death but partial suppression of PP2A Ca results in cell transformation. In this manuscript, we included two shRNAs targeting PP2A Ca. shPP2A Cα1 suppresses PP2A Ca completely and induces cell death while expression of shPP2A Cα1 results in partial suppression of shPP2A Ca and cell transformation.

In addition, we included two shRNAs targeting PP2A B56γ. Expression of shB56γ1 suppressed B56γ to a greater extent than found when we expressed shB56γ2 and correspondingly induced a stronger transformation phenotype. To clarify these findings, we now include mRNA levels for each of the targeted genes in Figure 1—figure supplement 1A-B. We note that the available antibodies specific for PP2A C recognize epitopes shared by PP2A Cα and Cβ, which makes it impossible to distinguish between these two isoforms by immunoblotting.

The data from GFP control (or the shLuc) are not represented because they have been used as controls to perform normalization (GFP control for ST, shLuc control for shRNAs against PP2A). We have added additional text to the Figure legends and text to clarify this experimental detail. In addition, we have added numbers to the heat map scale in the revised Figure 1.

2) A key conclusion of the previous version of this manuscript was MAP4K4-directed PP2A-STRN activity in the presence of ST. A direct experimental validation of this conclusion was requested. The revised manuscript includes this experiment, but as stated by the authors, the presence of ST in the PP2A complex could not be confirmed because of reagent issues (ST co-migrates with small IgG band). It is unclear how the data presented were normalized – to STRN4 inputs, or to co-IP-ed PP2A-C levels in the STRN4 IPs? The authors seem to have used equal amounts of total lysate in both conditions to IP STRN4 from, but this does not suffice to show that equal STRN4 levels were IP-ed, let alone that equal PP2A-C levels were present in both IPs. Could the ST levels in these IPs be determined by an MS-based approach? Could recombinant ST be added to STRN4 IPs and the effect on PP2A activity measured?

The reviewers have raised an important question. In the revised manuscript, we re-analyzed our mass spectrometry experiments to determine whether ST peptides were present in in the STRN4 immune complexes isolated from cells expressing ST but not GFP. Specifically, we used a newly available 2019 database of human proteins (vs. the 2015 database in the prior version of the manuscript) and also searched for methionine modifications. We found that methionine modified peptides corresponding to STare indeed present in these immune complexes. We now include this new analysis in Figure 6—figure supplement 1A in the revised manuscript.

The reviewers also requested clarification on whether our immunoprecipitation experiments were performed under conditions where equal amounts of STRN4 were loaded for the phosphatase assay in Figure 3E. We now provide IP/immunoblotting data to confirm that equal amounts of STRN4 was analyzed from GFP- and ST-expressing cells (Figure 3—figure supplement 1C).

The reviewers suggested an experiment to use recombinant ST to address this issue. In our experience, recombinant ST is difficult to produce, and several lines of evidence indicate that the assembly of PP2A complexes is a complex process that is incompletely understood. Since we were able to use mass spectrometry to answer the key question raised by the reviewers, we believe that the new data included in the revised manuscript more directly addresses this question.

3) In Figure 6B, more PP2A-C is co-IPed with STRN4 in ST overexpressing cells, while in Figure 3—figure supplement 1B this does not seem to be the case (in Figure 3C it is more difficult to judge). If more PP2A-C is co-IPed with STRN4 in ST versus GFP overexpressing cells, this would provide evidence for a relative redistribution of PP2A-C subunits into STRN4 complexes versus other PP2A B subunit-type complexes upon ST expression, and provide an explanation for the observed increase in activity. If, however, this would not be the case, and the presence of ST can be demonstrated in the complex, this would suggest that ST might act as 'an activator' of STRN4-PP2A-mediated dephosphorylation of certain MAP4K4 sites. These questions lead to a lack of clarity concerning the mechanism-of-action of how ST stimulates STRN4-PP2A activity towards MAP4K4.

We thank the reviewers for identifying this apparent discrepancy. To address this question, we have repeated these experiments several times to assess the amount of PP2A C associated with STRN4 in ST-expressing cells. We have confirmed that in each case, we found no difference in the total amount of PP2A C subunit that is associated with STRN4 in ST- relative to GFP-expressing cells, as shown in the Figure 3C and Figure 3—figure supplement 1C. We have revised Figure 3C, Figure 3—figure supplement 1B, Figure 6C, as well as Figure 1—figure supplement 1B, Figure 6—figure supplement 1B to indicate that we are assessing total PP2A C subunit expression, rather than only the PP2A Cα subunit in the immunoblots.

While we originally observed 2.7-fold increase in Cα binding to STRN4 in ST- relative to GFP-expressing cells as shown in Figure 6B, when we re-analyzed the proteomic results to address reviewer’s point #2 using the latest database (2019 Human Database), we observed this interaction to be more modest (1.7-fold) than originally observed. We believe that this is because values used for the quantitation of the MudPIT results are relative values normalized against all other proteins in a run. As more proteins and spectral counts were retrieved in the re-analysis using the latest database (e.g. 4517 vs. 8064 peptides), many of the values for proteins changed.

In addition, we observed that binding of PP2A Cb to STRN4 only occurred in GFP- but not the ST-expressing cells. When we factor in the relative abundance of both PP2A C subunits, we find that the net changes in the interaction of STRN4 with the PP2A C subunits is 1.1-fold, which is consistent with the co-IP experiment where the antibody cannot distinguish Cα from Cβ but simply detects total PP2A C isoforms.

Gene*STRN4-IP_GFP*
*(Normalised to STRN4)**STRN4-IP_ST*
*(Normalised to STRN4)**Fold change (ST/GFP)**PPP2CA**0.30**0.52**1.71**PPP2CB**0.15**X**X**PP2A C (CA+Cb)**0.46**0.52**1.14*

Taken together, these observations and analyses suggest that ST promotes the dephosphorylation of specific *MAP4K4* sites (Figure 3E) without increasing the specific association of total C subunit with STRN4. To reflect this, we have revised Figure 6B to include STRN4 interactions with both PP2A catalytic C isoforms, as well as updated the results of the re-analysis. We have also added more text in the Discussion section of the manuscript to address this important point. Additional experiments beyond the scope of this work are required to fully elucidate how ST regulates the associated PP2A activity towards *MAP4K4*.

4) Figure 6—figure supplement 1C: indications of statistical significance are missing

We have performed a student’s t-test and added the p-values to the revised Figure 6—figure supplement 1D.